# Supervision Complexity and its Role in Knowledge Distillation

**Hrayr Harutyunyan**[*1]    **Ankit Singh Rawat**[2]    **Aditya Krishna Menon**[2]
**Seungyeon Kim**[2]    **Sanjiv Kumar**[2]
[1] USC Information Sciences Institute    [2] Google Research NYC
`hrayrhar@usc.edu, {ankitsrawat,adityakmenon,seungyeonk,sanjivk}@google.com`

## Abstract

Despite the popularity and efficacy of knowledge distillation, there is limited understanding of why it helps. In order to study the generalization behavior of a distilled student, we propose a new theoretical framework that leverages *supervision complexity*: a measure of alignment between teacher-provided supervision and the student's *neural tangent kernel*. The framework highlights a delicate interplay among the teacher's accuracy, the student's margin with respect to the teacher predictions, and the complexity of the teacher predictions. Specifically, it provides a rigorous justification for the utility of various techniques that are prevalent in the context of distillation, such as early stopping and temperature scaling. Our analysis further suggests the use of *online distillation*, where a student receives increasingly more complex supervision from teachers in different stages of their training. We demonstrate efficacy of online distillation and validate the theoretical findings on a range of image classification benchmarks and model architectures.

## 1 Introduction

Knowledge distillation (KD) (Buciluǎ et al., 2006; Hinton et al., 2015) is a popular method of compressing a large "teacher" model into a more compact "student" model. In its most basic form, this involves training the student to fit the teacher's predicted *label distribution* or *soft labels* for each sample. There is strong empirical evidence that distilled students usually perform better than students trained on raw dataset labels (Hinton et al., 2015; Furlanello et al., 2018; Stanton et al., 2021; Gou et al., 2021). Multiple works have devised novel KD procedures that further improve the student model performance (see Gou et al. (2021) and references therein). Simultaneously, several works have aimed to rigorously formalize *why* KD can improve the student model performance. Some prominent observations from this line of work are that (self-)distillation induces certain favorable optimization biases in the training objective (Phuong & Lampert, 2019; Ji & Zhu, 2020), lowers variance of the objective (Menon et al., 2021; Dao et al., 2021; Ren et al., 2022), increases regularization towards learning "simpler" functions (Mobahi et al., 2020), transfers information from different data views (Allen-Zhu & Li, 2020), and scales per-example gradients based on the teacher's confidence (Furlanello et al., 2018; Tang et al., 2020).

Despite this remarkable progress, there are still many open problems and unexplained phenomena around knowledge distillation; to name a few:

— *Why do soft labels (sometimes) help?* It is agreed that teacher's soft predictions carry information about class similarities (Hinton et al., 2015; Furlanello et al., 2018), and that this softness of predictions has a regularization effect similar to label smoothing (Yuan et al., 2020). Nevertheless, KD also works in binary classification settings with limited class similarity information (Müller et al., 2020). How exactly the softness of teacher predictions (controlled by a temperature parameter) affects the student learning remains far from well understood.

— *The role of capacity gap*. There is evidence that when there is a significant capacity gap between the teacher and the student, the distilled model usually falls behind its teacher (Mirzadeh

---

[*]Work done while interning at Google Research NYC.

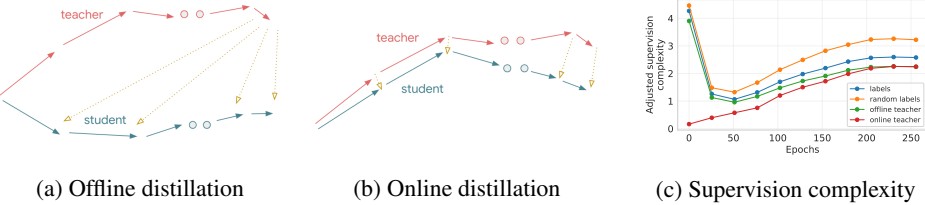

| (a) Offline distillation | (b) Online distillation | (c) Supervision complexity |

Figure 1: Online vs. online distillation. Figures (a) and (b) illustrate possible teacher and student function trajectories in offline and offline KD. The yellow dotted lines indicate KD. Figure (c) plots adjusted supervision complexity of various targets with respect to NTKs at different stages of training (see §4 for more details).

et al., 2020; Cho & Hariharan, 2019; Stanton et al., 2021). It is unclear whether this is due to difficulties in optimization, or due to insufficient student capacity.

— *What makes a good teacher?* Sometimes less accurate models are better teachers (Cho & Hariharan, 2019; Mirzadeh et al., 2020). Moreover, early stopped or exponentially averaged models are often better teachers (Ren et al., 2022). A comprehensive explanation of this remains elusive.

The aforementioned wide range of phenomena suggest that there is a complex interplay between teacher accuracy, softness of teacher-provided targets, and complexity of the distillation objective.

This paper provides a new theoretically grounded perspective on KD through the lens of *supervision complexity*. In a nutshell, this quantifies why certain targets (e.g., temperature-scaled teacher probabilities) may be "easier" for a student model to learn compared to others (e.g., raw one-hot labels), owing to better alignment with the student's *neural tangent kernel* (*NTK*) (Jacot et al., 2018; Lee et al., 2019). In particular, we provide a novel theoretical analysis (§2, Thm. 3 and 4) of the role of supervision complexity on kernel classifier generalization, and use this to derive a new generalization bound for distillation (Prop. 5). The latter highlights how student generalization is controlled by a balance of the *teacher generalization*, the student's *margin* with respect to the teacher predictions, and the complexity of the teacher's predictions.

Based on the preceding analysis, we establish the conceptual and practical efficacy of a simple *online distillation* approach (§4), wherein the student is fit to progressively more complex targets, in the form of teacher predictions at various checkpoints during its training. This method can be seen as guiding the student in the function space (see Fig. 1), and leads to better generalization compared to offline distillation. We provide empirical results on a range of image classification benchmarks confirming the value of online distillation, particularly for students with weak inductive biases.

Beyond practical benefits, the supervision complexity view yields new insights into distillation:

— *The role of temperature scaling and early-stopping.* Temperature scaling and early-stopping of the teacher have proven effective for KD. We show that both of these techniques reduce the supervision complexity, at the expense of also lowering the classification margin. Online distillation manages to smoothly increase teacher complexity, without degrading the margin.

— *Teaching a weak student.* We show that for students with weak inductive biases, and/or with much less capacity than the teacher, the final teacher predictions *are often as complex as dataset labels*, particularly during the early stages of training. In contrast, online distillation allows the supervision complexity to progressively increase, thus allowing even a weak student to learn.

— *NTK and relational transfer.* We show that online distillation is highly effective at matching the teacher and student NTK matrices. This transfers *relational knowledge* in the form of example-pair similarity, as opposed to standard distillation which only transfers *per-example knowledge*.

**Problem setting.** We focus on classification problems from input domain $\mathcal{X}$ to $d$ classes. We are given a training set of $n$ labeled examples $\{(x_1, y_1), \ldots, (x_n, y_n)\}$, with one-hot encoded labels $y_i \in \{0, 1\}^d$. Typically, a model $f_\theta : \mathcal{X} \to \mathbb{R}^d$ is trained with the *softmax cross-entropy* loss:

$$\mathcal{L}_{\mathrm{ce}}(f_\theta) = -\frac{1}{n} \sum\nolimits_{i=1}^{n} y_i^\top \log \sigma(f_\theta(x_i)), \tag{1}$$

where $\sigma(\cdot)$ is the softmax function. In standard KD, given a trained *teacher* model $g : \mathcal{X} \to \mathbb{R}^d$ that outputs logits, one trains a *student* model $f_\theta : \mathcal{X} \to \mathbb{R}^d$ to fit the teacher predictions. Hinton et al. (2015) propose the following KD loss:

$$\mathcal{L}_{\text{kd-ce}}(f_\theta; g, \tau) = -\frac{\tau^2}{n} \sum\nolimits_{i=1}^{n} \sigma(g(x_i)/\tau)^\top \log \sigma(f_\theta(x_i)/\tau), \tag{2}$$

where temperature $\tau > 0$ controls the softness of teacher predictions. To highlight the effect of KD and simplify exposition, we assume that the student is not trained with the dataset labels.

## 2 SUPERVISION COMPLEXITY AND GENERALIZATION

One apparent difference between standard training and KD (Eq. 1 and 2) is that the latter modifies the *targets* that the student attempts to fit. The targets used during distillation ensure a better generalization for the student; what is the reason for this? Towards answering this question, we present a new perspective on KD in terms of *supervision complexity*. To begin, we show how the generalization of a *kernel-based* classifier is controlled by a measure of alignment between the target labels and the kernel matrix. We first treat *binary* kernel-based classifiers (Thm. 3), and later extend our analysis to *multiclass* kernel-based classifiers (Thm. 4). Finally, by leveraging the neural tangent kernel machinery, we discuss the implications of our analysis for neural classifiers in §2.2.

### 2.1 SUPERVISION COMPLEXITY CONTROLS KERNEL MACHINE GENERALIZATION

The notion of supervision complexity is easiest to introduce and study for kernel-based classifiers. We briefly review some necessary background (Scholkopf & Smola, 2001). Let $k : \mathcal{X} \times \mathcal{X} \to \mathbb{R}$ be a positive semidefinite kernel defined over an input space $\mathcal{X}$. Any such kernel uniquely determines a reproducing kernel Hilbert space (RKHS) $\mathcal{H}$ of functions from $\mathcal{X}$ to $\mathbb{R}$. This RKHS is the completion of the set of functions of form $f(x) = \sum_{i=1}^{m} \alpha_i k(x_i, x)$, with $x_i \in \mathcal{X}, \alpha_i \in \mathbb{R}$. Any $f(x) = \sum_{i=1}^{m} \alpha_i k(x_i, x) \in \mathcal{H}$ has (RKHS) norm $\|f\|_{\mathcal{H}}^2 = \sum_{i=1}^{m} \sum_{j=1}^{m} \alpha_i \alpha_j k(x_i, x_j) = \boldsymbol{\alpha}^\top K \boldsymbol{\alpha}$, where $\boldsymbol{\alpha} = (\alpha_1, \ldots, \alpha_n)^\top$ and $K_{i,j} = k(x_i, x_j)$. Intuitively, $\|f\|_{\mathcal{H}}^2$ measures the *smoothness* of $f$, e.g., for a Gaussian kernel it measures the Fourier spectrum decay of $f$ (Scholkopf & Smola, 2001).

For simplicity, we start with the case of binary classification. Suppose $\{(X_i, Y_i)\}_{i \in [n]}$ are $n$ i.i.d. examples sampled from some probability distribution on $\mathcal{X} \times \mathcal{Y}$, with $\mathcal{Y} \subset \mathbb{R}$, where positive and negative labels correspond to distinct classes. Let $K_{i,j} = k(X_i, X_j)$ denote the kernel matrix, and $\boldsymbol{Y} = (Y_1, \ldots, Y_n)^\top$ be the concatenation of all training labels.

**Definition 1** (Supervision complexity)**.** The *supervision complexity* of targets $Y_1, \ldots, Y_n$ with respect to a kernel $k$ is defined to be $\boldsymbol{Y}^\top K^{-1} \boldsymbol{Y}$ in cases when $K$ is invertible, and $+\infty$ otherwise.

We now establish how supervision complexity controls the *smoothness* of the optimal kernel classifier. Consider a classifier obtained by solving a *regularized* kernel classification problem:

$$f^* \in \operatorname*{argmin}_{f \in \mathcal{H}} \frac{1}{n} \sum\nolimits_{i=1}^{n} \ell(f(X_i), Y_i) + \frac{\lambda}{2} \|f\|_{\mathcal{H}}^2, \tag{3}$$

where $\ell$ is a loss function and $\lambda > 0$. The following proposition shows whenever the supervision complexity is small, the RKHS norm of any optimal solution $f^*$ will also be small (see Appendix B for a proof). This is an important learning bias that shall help us explain certain aspects of KD.

**Proposition 2.** *Assume that $K$ is full rank almost surely; $\ell(y, y') \geq 0, \forall y, y' \in \mathcal{Y}$; and $\ell(y, y) = 0, \forall y \in \mathcal{Y}$. Then, with probability 1, for any solution $f^*$ of (3), we have $\|f^*\|_{\mathcal{H}}^2 \leq \boldsymbol{Y}^\top K^{-1} \boldsymbol{Y}$.*

Equipped with the above result, we now show how supervision complexity controls generalization. In the following, let $\phi_\gamma : \mathbb{R} \to [0, 1]$ be the *margin loss* (Mohri et al., 2018) with scale $\gamma > 0$:

$$\phi_\gamma(\alpha) = \min\{1, \max\{1 - \alpha/\gamma, 0\}\}. \tag{4}$$

**Theorem 3.** *Assume that $\kappa = \sup_{x \in \mathcal{X}} k(x, x) < \infty$ and $K$ is full rank almost surely. Further, assume that $\ell(y, y') \geq 0, \forall y, y' \in \mathcal{Y}$ and $\ell(y, y) = 0, \forall y \in \mathcal{Y}$. Let $M_0 = \lceil \gamma \sqrt{n}/(2\sqrt{\kappa}) \rceil$. Then, with probability at least $1 - \delta$, for any solution $f^*$ of problem in Eq. (3), we have*

$$\mathbb{P}_{X,Y}(Y f^*(X) \leq 0) \leq \frac{1}{n} \sum_{i=1}^{n} \phi_\gamma(\operatorname{sign}(Y_i) f^*(X_i)) + \frac{2\sqrt{\boldsymbol{Y}^\top K^{-1} \boldsymbol{Y}} + 2}{\gamma n} \sqrt{\operatorname{Tr}(K)} + 3\sqrt{\frac{\ln(2M_0/\delta)}{2n}}. \tag{5}$$

The proof is available in Appendix B. One can compare Thm. 3 with the standard Rademacher bound for kernel classifiers (Bartlett & Mendelson, 2002). The latter typically consider learning over functions with RKHS norm bounded by a *constant* $M > 0$. The corresponding complexity term then decays as $\mathcal{O}(\sqrt{M \cdot \text{Tr}(K)/n})$, which is *data-independent*. Consequently, such a bound cannot adapt to the intrinsic "difficulty" of the targets $\boldsymbol{Y}$. In contrast, Thm. 3 considers functions with RKHS norm bounded by the *data-dependent* supervision complexity term. This results in a more informative generalization bound, which captures the "difficulty" of the targets. Here, we note that Arora et al. (2019) characterized the generalization of an overparameterized two-layer neural network via a term closely related to the supervision complexity (see §5 for additional discussion).

The supervision complexity $\boldsymbol{Y}^\top K^{-1} \boldsymbol{Y}$ is small whenever $\boldsymbol{Y}$ is aligned with top eigenvectors of $K$ and/or $\boldsymbol{Y}$ has small scale. Furthermore, one cannot make the bound close to zero by just reducing the scale of targets, as one would need a small $\gamma$ to control the margin loss that would otherwise increase due to student predictions getting closer to zero (as the student aims to match $Y_i$).

To better understand the role of supervision complexity, it is instructive to consider two special cases that lead to a poor generalization bound: (1) uninformative *features*, and (2) uninformative *labels*.

**Complexity under uninformative features.** Suppose the kernel matrix $K$ is diagonal, so that the kernel provides *no* information on example-pair similarity; i.e., the kernel is "uninformative". An application of Cauchy-Schwarz reveals the key expression in the second term in (5) satisfies:

$$\frac{1}{n}\sqrt{\boldsymbol{Y}^\top K^{-1}\boldsymbol{Y}\,\text{Tr}(K)} = \frac{1}{n}\sqrt{\left(\sum_{i=1}^n Y_i^2 \cdot k(X_i, X_i)^{-1}\right)\left(\sum_{i=1}^n k(X_i, X_i)\right)} \geq \frac{1}{n}\sum_{i=1}^n |Y_i|.$$

Consequently, this term is least constant in order, and *does not* vanish as $n \to \infty$.

**Complexity under uninformative labels.** Suppose the labels $Y_i$ are purely random, and independent from inputs $X_i$. Conditioned on $\{X_i\}$, $\boldsymbol{Y}^\top K^{-1}\boldsymbol{Y}$ concentrates around its mean by the Hanson-Wright inequality (Vershynin, 2018). Hence, $\exists \epsilon(K, \delta, n)$ such that with probability $\geq 1 - \delta$, $\boldsymbol{Y}^\top K^{-1}\boldsymbol{Y} \geq \mathbb{E}_{\{Y_i\}}\left[\boldsymbol{Y}^\top K^{-1}\boldsymbol{Y}\right] - \epsilon = \mathbb{E}\left[Y_1^2\right]\text{Tr}(K^{-1}) - \epsilon$. Thus, with the same probability,

$$\frac{1}{n}\sqrt{\boldsymbol{Y}^\top K^{-1}\boldsymbol{Y}\,\text{Tr}(K)} \geq \frac{1}{n}\sqrt{(\mathbb{E}\left[Y_1^2\right]\text{Tr}(K^{-1}) - \epsilon)\,\text{Tr}(K)} \geq \frac{1}{n}\sqrt{\mathbb{E}\left[Y_1^2\right]n^2 - \epsilon\,\text{Tr}(K)},$$

where the last inequality is by Cauchy-Schwarz. For sufficiently large $n$, the quantity $\mathbb{E}\left[Y_1^2\right]n^2$ dominates $\epsilon\,\text{Tr}(K)$, rendering the bound of Thm. 3 close to a constant.

## 2.2 EXTENSIONS: MULTICLASS CLASSIFICATION AND NEURAL NETWORKS

We now show that a result similar to Thm. 3 holds for multiclass classification as well. In addition, we also discuss how our results are instructive about the behavior of neural networks.

**Extension to multiclass classification.** Let $\{(X_i, Y_i)\}_{i \in [n]}$ be drawn i.i.d. from a distribution over $\mathcal{X} \times \mathcal{Y}$, where $\mathcal{Y} \subset \mathbb{R}^d$. Let $k : \mathcal{X} \times \mathcal{X} \to \mathbb{R}^{d \times d}$ be a matrix-valued positive definite kernel and $\mathcal{H}$ be the corresponding vector-valued RKHS. As in the binary classification case, we consider a kernel problem in Eq. (3). Let $\boldsymbol{Y}^\top = (Y_1^\top, \ldots, Y_n^\top)$ and $K$ be the kernel matrix of training examples:

$$K = \begin{bmatrix} k(X_1, X_1) & \cdots & k(X_1, X_n) \\ \cdots & \cdots & \cdots \\ k(X_n, X_1) & \cdots & k(X_n, X_n) \end{bmatrix} \in \mathbb{R}^{nd \times nd}. \tag{6}$$

For $f : \mathcal{X} \to \mathbb{R}^d$ and a labeled example $(x, y)$, let $\rho_f(x, y) = f(x)_y - \max_{y' \neq y} f(x)_{y'}$ be the *prediction margin*. Then, the following analogue of Thm. 3 holds (see Appendix C for the proof).

**Theorem 4.** *Assume that* $\kappa = \sup_{x \in \mathcal{X}, y \in [d]} k(x, x)_{y,y} < \infty$, *and* $K$ *is full rank almost surely. Further, assume that* $\ell(y, y') \geq 0, \forall y, y' \in \mathcal{Y}$ *and* $\ell(y, y) = 0, \forall y \in \mathcal{Y}$. *Let* $M_0 = \lceil \gamma\sqrt{n}/(4d\sqrt{\kappa}) \rceil$. *Then, with probability at least* $1 - \delta$, *for any solution* $f^*$ *of problem in Eq. (3),*

$$\mathbb{P}_{X,Y}(\rho_{f^*}(X, Y) \leq 0) \leq \frac{1}{n}\sum_{i=1}^n \mathbf{1}\{\rho_{f^*}(X_i, Y_i) \leq \gamma\} + \frac{4d(\boldsymbol{Y}^\top K^{-1}\boldsymbol{Y} + 1)}{\gamma n}\sqrt{\text{Tr}(K)} + 3\sqrt{\frac{\log(2M_0/\delta)}{2n}}. \tag{7}$$

**Implications for neural classifiers**. Our analysis has so far focused on kernel-based classifiers. While neural networks are not exactly kernel methods, many aspects of their performance can be understood via a corresponding *linearized neural network* (see Ortiz-Jiménez et al. (2021) and references therein). We follow this approach, and given a neural network $f_\theta$ with current weights $\theta_0$, we consider the corresponding linearized neural network comprising the linear terms of the Taylor expansion of $f_\theta(x)$ around $\theta_0$ (Jacot et al., 2018; Lee et al., 2019): $f_\theta^{\text{lin}}(x) \triangleq f_{\theta_0}(x) + \nabla_\theta f_{\theta_0}(x)^\top (\theta - \theta_0)$. Let $\omega \triangleq \theta - \theta_0$. This network $f_\omega^{\text{lin}}(x)$ is a linear function with respect to the parameters $\omega$, but is generally non-linear with respect to the input $x$. Note that $\nabla_\theta f_{\theta_0}(x)$ acts as a feature representation, and induces the *neural tangent kernel (NTK)* $k_0(x, x') = \nabla_\theta f_{\theta_0}(x)^\top \nabla_\theta f_{\theta_0}(x') \in \mathbb{R}^{d \times d}$.

Given a labeled dataset $S = \{(x_i, y_i)\}_{i \in [n]}$ and a loss function $\mathcal{L}(f; S)$, the dynamics of gradient flow with learning rate $\eta > 0$ for $f_\omega^{\text{lin}}$ can be fully characterized in the function space, and depends only on the predictions at $\theta_0$ and the NTK $k_0$:

$$\dot{f}_t^{\text{lin}}(x') = -\eta \cdot K_0(x', x_{1:n}) \ \nabla_f \mathcal{L}(f_t^{\text{lin}}(x_{1:n}); S), \tag{8}$$

where $f(x_{1:n}) \in \mathbb{R}^{nd}$ denotes the concatenation of predictions on training examples and $K_0(x', x_{1:n}) = \nabla_\theta f_{\theta_0}(x')^\top \nabla_\theta f_{\theta_0}(x_{1:n})$. Lee et al. (2019) show that as one increases the width of the network or when $(\theta - \theta_0)$ does not change much during training, the dynamics of the linearized and original neural network become close.

When $f_\theta$ is sufficiently overparameterized and $\mathcal{L}$ is convex with respect to $\omega$, then $\omega_t$ converges to an interpolating solution. Furthermore, for the mean squared error objective, the solution has the minimum Euclidean norm (Gunasekar et al., 2017). As the Euclidean norm of $\omega$ corresponds to norm of $f_\omega^{\text{lin}}(x) - f_{\theta_0}(x)$ in the vector-valued RKHS $\mathcal{H}$ corresponding to $k_0$, training a linearized network to interpolation is equivalent to solving the following with a small $\lambda > 0$:

$$h^* = \underset{h \in \mathcal{H}}{\operatorname{argmin}} \frac{1}{n} \sum_{i=1}^n (f_{\theta_0}(x_i) + h(x_i) - y_i)^2 + \frac{\lambda}{2} \|h\|_\mathcal{H}^2. \tag{9}$$

Therefore, the generalization bounds of Thm. 3 and 4 apply to $h^*$ with supervision complexity of residual targets $y_i - f_{\theta_0}(x_i)$. However, we are interested in the performance of $f_{\theta_0} + h^*$. As the proofs of these results rely on bounding the Rademacher complexity of hypothesis sets of form $\{h \in \mathcal{H} \colon \|h\| \leq M\}$, and shifting a hypothesis set by a constant function does not change the Rademacher complexity (see Remark 7), these proofs can be easily modified to handle hypotheses shifted by the constant function $f_{\theta_0}$.

## 3 KNOWLEDGE DISTILLATION: A SUPERVISION COMPLEXITY LENS

We now turn to KD, and explore how supervision complexity affects student's generalization. We show that student's generalization depends on three terms: the *teacher generalization*, the student's *margin* with respect to the teacher predictions, and the complexity of the teacher's predictions.

### 3.1 TRADE-OFF BETWEEN TEACHER ACCURACY, MARGIN, AND COMPLEXITY

Consider the binary classification setting of §2, and a fixed teacher $g : \mathcal{X} \to \mathbb{R}$ that outputs a logit. Let $\{(X_i, Y_i^*)\}_{i \in [n]}$ be $n$ i.i.d. labeled examples, where $Y_i^* \in \{-1, 1\}$ denotes the ground truth labels. For temperature $\tau > 0$, let $Y_i \triangleq 2\,\sigma(g(X_i)/\tau) - 1 \in [-1, +1]$ denote the teacher's *soft predictions*, for sigmoid function $\sigma \colon z \mapsto (1 + \exp(-z))^{-1}$. Our key observation is: if the teacher predictions $Y_i$ are accurate enough and have significantly lower complexity compared to ground truth labels $Y_i^*$, then a student kernel method (cf. Eq. (3)) trained with $Y_i$ can generalize better than the one trained with $Y_i^*$. The following result quantifies the trade-off between teacher accuracy, student prediction margin, and teacher prediction complexity (see Appendix B.3 for a proof).

**Proposition 5.** *Assume that $\kappa = \sup_{x \in \mathcal{X}} k(x, x) < \infty$ and $K$ is full rank almost surely, $\ell(y, y') \geq 0, \forall y, y' \in \mathcal{Y}$, and $\ell(y, y) = 0, \forall y \in \mathcal{Y}$. Let $Y_i$ and $Y_i^*$ be defined as above. Let*

$M_0 = \lceil \gamma \sqrt{n}/(2\sqrt{\kappa}) \rceil$. *Then, with probability at least* $1 - \delta$, *any solution* $f^*$ *of problem* (3) *satisfies*

$$\underbrace{\mathbb{P}_{X,Y^*}(Y^* f^*(X) \leq 0)}_{student\ risk} \leq \underbrace{\mathbb{P}_{X,Y^*}(Y^* g(X) \leq 0)}_{teacher\ risk} + \underbrace{\frac{1}{n}\sum_{i=1}^{n} \phi_\gamma\left(\text{sign}\left(Y_i\right) f^*(X_i)\right)}_{student's\ empirical\ margin\ loss\ w.r.t.\ teacher\ predictions}$$

$$+ \underbrace{\left(2\sqrt{\boldsymbol{Y}^\top K^{-1}\boldsymbol{Y}} + 2\right)\sqrt{\text{Tr}\left(K\right)}/(\gamma n)}_{complexity\ of\ teacher's\ predictions} + 3\sqrt{\ln\left(2M_0/\delta\right)/(2n)}.$$

Note that a similar result is easy to establish for multiclass classification using Thm. 4. The first term in the above accounts for the misclassification rate of the teacher. While this term is not irreducible (it is possible for a student to perform better than its teacher), generally a student performs worse that its teacher, especially when there is a significant teacher-student capacity gap. The second term is student's empirical margin loss w.r.t. teacher predictions. This captures the price of making teacher predictions too soft. Intuitively, the softer (i.e., closer to zero) teacher predictions are, the harder it is for the student to learn the classification rule. The third term accounts for the supervision complexity and the margin parameter $\gamma$. Thus, one has to choose $\gamma$ carefully to achieve a good balance between empirical margin loss and margin-normalized supervision complexity.

**The effect of temperature.** For a fixed margin parameter $\gamma > 0$, increasing the temperature $\tau$ makes teacher's predictions $Y_i$ softer. On the one hand, the reduced scale decreases the supervision complexity $\boldsymbol{Y}^\top K^{-1}\boldsymbol{Y}$. Moreover, we shall see that in the case of neural networks the complexity decreases even further due to $\boldsymbol{Y}$ becoming more aligned with top eigenvectors of $K$. On the other hand, the scale of predictions of the (possibly interpolating) student $f^*$ will decrease too, increasing the empirical margin loss. This suggests that setting the value of $\tau$ is not trivial: the optimal value can be different based on the kernel $k$ and teacher logits $g(X_i)$.

## 3.2 FROM OFFLINE TO ONLINE KNOWLEDGE DISTILLATION

We identified that supervision complexity plays a key role in determining the efficacy of a distillation procedure. The supervision from a fully trained teacher model can prove to be very complex for a student model in an early stage of its training (Fig. 1c). This raises the question: *is there value in providing progressively difficult supervision to the student?* In this section, we describe a simple online distillation method, where the the teacher is updated during the student training.

Over the course of their training, neural models learn functions of increasing complexity (Kalimeris et al., 2019). This provides a natural way to construct a set of teachers with varying prediction complexities. Similar to Jin et al. (2019), for practical considerations of not training the teacher and the student simultaneously, we assume the availability of teacher checkpoints over the course of its training. Given $m$ teacher checkpoints at times $\mathcal{T} = \{t_i\}_{i \in [m]}$, during the $t$-th step of distillation, the student receives supervision from the teacher checkpoint at time $\min\{t' \in \mathcal{T} : t' > t\}$. Note that the student is trained for the same number of epochs in total as in offline distillation. Throughout the paper we use the term "online distillation" for this approach (cf. Algorithm 1 of Appendix A).

Online distillation can be seen as guiding the student network to follow the teacher's trajectory in *function space* (see Fig. 1). Given that NTK can be interpreted as a principled notion of example similarity and controls which examples affect each other during training (Charpiat et al., 2019), it is desirable for the student to have an NTK similar to that of its teacher at each time step. To test whether online distillation also transfers NTK, we propose to measure similarity between the final student and final teacher NTKs. For computational efficiency we work with NTK matrices corresponding to a batch of $b$ examples ($bd \times bd$ matrices). Explicit computation of even batch NTK matrices can be costly, especially when the number of classes $d$ is large. We propose to view student and teacher batch NTK matrices (denoted by $K_f$ and $K_g$ respectively) as operators and measure their similarity by comparing their behavior on random vectors: $\text{sim}(K_f, K_g) = \mathbb{E}_{v \sim \mathcal{N}(0,I)}\left[\langle K_f v, K_g v \rangle / \left(\|K_f v\| \|K_g v\|\right)\right]$. Note that the cosine distance is used to account for scale differences of $K_g$ and $K_f$. The kernel-vector products appearing in this similarity measure above can be computed efficiently without explicitly constructing the kernel matrices. For example, $K_f v = \nabla_\theta f_\theta(x_{1:b})^\top \left(\nabla_\theta f_\theta(x_{1:b})v\right)$ can be computed with one vector-Jacobian product followed by a Jacobian-vector product. The former can be computed efficiently using backpropagation, while the latter can be computed efficiently using forward-mode differentiation.

Table 1: Results on CIFAR-100.

| Setting | No KD | Offline KD | | Online KD | | Teacher |
|---|---|---|---|---|---|---|
| | | $\tau = 1$ | $\tau = 4$ | $\tau = 1$ | $\tau = 4$ | |
| ResNet-56 → LeNet-5x8 | $47.3 \pm 0.6$ | $50.1 \pm 0.4$ | $59.9 \pm 0.2$ | $61.9 \pm 0.2$ | $66.1 \pm 0.4$ | 72.0 |
| ResNet-56 → ResNet-20 | $67.7 \pm 0.5$ | $68.2 \pm 0.3$ | $71.6 \pm 0.2$ | $69.6 \pm 0.3$ | $71.4 \pm 0.3$ | 72.0 |
| ResNet-110 → LeNet-5x8 | $47.2 \pm 0.5$ | $48.6 \pm 0.8$ | $59.0 \pm 0.3$ | $60.8 \pm 0.2$ | $65.8 \pm 0.2$ | 73.4 |
| ResNet-110 → ResNet-20 | $67.8 \pm 0.3$ | $67.8 \pm 0.2$ | $71.2 \pm 0.0$ | $69.0 \pm 0.3$ | $71.4 \pm 0.0$ | 73.4 |

Table 2: Results on Tiny ImageNet.

| Setting | No KD | Offline KD | | Online KD | | Teacher |
|---|---|---|---|---|---|---|
| | | $\tau = 1$ | $\tau = 2$ | $\tau = 1$ | $\tau = 2$ | |
| MobileNet-V3-125 → MobileNet-V3-35 | $58.5 \pm 0.2$ | $59.2 \pm 0.1$ | $60.2 \pm 0.2$ | $60.7 \pm 0.2$ | $62.3 \pm 0.3$ | 62.7 |
| ResNet-101 → MobileNet-V3-35 | $58.5 \pm 0.2$ | $59.4 \pm 0.5$ | $61.6 \pm 0.2$ | $61.1 \pm 0.3$ | $62.0 \pm 0.3$ | 66.0 |
| MobileNet-V3-125 → VGG-16 | $48.9 \pm 0.3$ | $54.1 \pm 0.4$ | $59.4 \pm 0.4$ | $58.9 \pm 0.7$ | $62.3 \pm 0.3$ | 62.7 |
| ResNet-101 → VGG-16 | $48.6 \pm 0.4$ | $53.1 \pm 0.4$ | $60.6 \pm 0.2$ | $60.4 \pm 0.2$ | $64.0 \pm 0.1$ | 66.0 |

## 4 EXPERIMENTAL RESULTS

We now present experimental results to showcase the importance of supervision complexity in distillation, and to establish efficacy of online distillation.

### 4.1 EXPERIMENTAL SETUP

We consider standard image classification benchmarks: CIFAR-10, CIFAR-100, and Tiny-ImageNet. Additionally, we derive a binary classification task from CIFAR-100 by grouping the first and last 50 classes into two meta-classes. We consider teacher and student architectures that are ResNets (He et al., 2016), VGGs (Simonyan & Zisserman, 2015), and MobileNets (Howard et al., 2019) of various depths. As a student architecture with relatively weaker inductive biases, we also consider the LeNet-5 (LeCun et al., 1998) with 8 times wider hidden layers. We use standard hyperparameters (Appendix A) to train these models. We compare (1) regular one-hot training (without any distillation), (2) regular offline distillation using the temperature-scaled softmax cross-entropy, and (3) online distillation using the same loss. For CIFAR-10 and binary CIFAR-100, we also consider training with mean-squared error (MSE) loss and its corresponding KD loss:

$$\mathcal{L}_{\text{mse}}(f_\theta) = \frac{1}{2n} \sum_{i=1}^{n} \|y_i - f_\theta(x_i)\|_2^2, \quad \mathcal{L}_{\text{kd-mse}}(f_\theta; g, \tau) = \frac{\tau}{2n} \sum_{i=1}^{n} \|\sigma(g(x_i)/\tau) - f_\theta(x_i)\|_2^2. \quad (10)$$

The MSE loss allows for interpolation in case of one-hot labels $y_i$, making it amenable to the analysis in §2 and §3. Moreover, Hui & Belkin (2021) show that under standard training, the CE and MSE losses perform similarly; as we shall see, the same is true for distillation as well.

As mentioned in Sec. 1, in all KD experiments student networks receive supervision only through a knowledge distillation loss (i.e., dataset labels are not used). This choice help us decrease differences between the theory and experiments. Furthermore, in our preliminary experiments we observed that this choice does not result in student performance degradation (see Appendix D).

### 4.2 RESULTS AND DISCUSSION

Tables 1, 2, 3 and Table 5 (Appendix D) present the results (mean and standard deviation of test accuracy over 3 random trials). First, we see that online distillation with proper temperature scaling typically yields the most accurate student. The gains over regular distillation are particularly pronounced when there is a large teacher-student gap. For example, on CIFAR-100, ResNet to LeNet distillation with temperature scaling appears to hit a limit of $\sim 60\%$ accuracy. Online distillation however manages to further increase accuracy by $+6\%$, which is a $\sim 20\%$ increase compared to standard training. Second, the similar results on binary CIFAR-100 shows that "dark knowledge" in the form of membership information in multiple classes is not necessary for distillation to succeed.

The results also demonstrate that knowledge distillation with the MSE loss of (10) has a qualitatively similar behavior to KD with CE objective. We use these MSE models to highlight the role of

Table 3: Results on binary CIFAR-100. Every second line is an MSE student.

| Setting | No KD | Offline KD | | Online KD | | Teacher |
|---|---|---|---|---|---|---|
| | | $\tau = 1$ | $\tau = 4$ | $\tau = 1$ | $\tau = 4$ | |
| ResNet-56 $\rightarrow$ LeNet-5x8 | $71.5 \pm 0.2$ | $72.4 \pm 0.1$ | $73.6 \pm 0.2$ | $74.7 \pm 0.2$ | $76.1 \pm 0.2$ | $77.9$ |
| ResNet-56 $\rightarrow$ LeNet-5x8 | $71.5 \pm 0.4$ | $71.9 \pm 0.3$ | $73.0 \pm 0.3$ | $75.1 \pm 0.3$ | $75.1 \pm 0.1$ | $77.9$ |
| ResNet-56 $\rightarrow$ ResNet-20 | $75.8 \pm 0.5$ | $76.1 \pm 0.2$ | $77.1 \pm 0.6$ | $77.8 \pm 0.3$ | $78.1 \pm 0.1$ | $77.9$ |
| ResNet-56 $\rightarrow$ ResNet-20 | $76.1 \pm 0.5$ | $76.0 \pm 0.2$ | $77.4 \pm 0.3$ | $78.0 \pm 0.2$ | $78.4 \pm 0.3$ | $77.9$ |
| ResNet-110 $\rightarrow$ LeNet-5x8 | $71.4 \pm 0.4$ | $71.9 \pm 0.1$ | $72.9 \pm 0.3$ | $74.3 \pm 0.3$ | $75.4 \pm 0.3$ | $78.4$ |
| ResNet-110 $\rightarrow$ LeNet-5x8 | $71.6 \pm 0.2$ | $71.5 \pm 0.4$ | $72.6 \pm 0.4$ | $74.8 \pm 0.4$ | $74.6 \pm 0.2$ | $78.4$ |
| ResNet-110 $\rightarrow$ ResNet-20 | $76.0 \pm 0.3$ | $76.0 \pm 0.2$ | $77.0 \pm 0.1$ | $77.3 \pm 0.2$ | $78.0 \pm 0.4$ | $78.4$ |
| ResNet-110 $\rightarrow$ ResNet-20 | $76.1 \pm 0.2$ | $76.4 \pm 0.3$ | $77.6 \pm 0.3$ | $77.9 \pm 0.2$ | $78.1 \pm 0.1$ | $78.4$ |

(a) Normalized complexity      (b) Temperature effect      (c) Teacher averaging

Figure 2: Supervision complexity for various targets. *On the left:* Normalized adjusted supervision complexities of various targets with respect to a LeNet-5x8 network at different stages of its training. *In the middle:* The effect of temperature on the supervision complexity of an offline teacher for a LeNet-5x8 after training for 25 epochs. *On the right:* The effect of averaging teacher predictions.

supervision complexity. As an instructive case, we consider a LeNet-5x8 network trained on binary CIFAR-100 with the standard MSE loss function. For a given checkpoint of this network and a given set of $m$ labeled (*test*) examples $\{(X_i, Y_i)\}_{i \in [m]}$, we compute the *adjusted supervision complexity* $1/n \sqrt{(\boldsymbol{Y} - f(X_{1:m}))^\top K^{-1} (\boldsymbol{Y} - f(X_{1:m})) \cdot \mathrm{Tr}(K)}$, where $f$ denotes the current prediction function, and $K$ is derived from the current NTK. Note that the subtraction of initial predictions is the appropriate way to measure complexity given the form of the optimization problem (9). As the training NTK matrix becomes aligned with dataset labels during training (see Baratin et al. (2021) and Fig. 7 of Appendix D), we pick $\{X_i\}_{i \in m}$ to be a set of $2^{12}$ *test* examples.

**Comparison of supervision complexities.** We compare the adjusted supervision complexities of random labels, dataset labels, and predictions of an offline and online ResNet-56 teacher predictions with respect to various checkpoints of the LeNet-5x8 network. The results presented in Fig. 1c indicate that the dataset labels and offline teacher predictions are as complex as random labels in the beginning. After some initial decline, the complexity of these targets increases as the network starts to overfit. Given the lower bound on the supervision complexity of random labels (see §2), this increase means that the NTK spectrum becomes less uniform (see Fig. 6 of Appendix D).

In contrast to these static targets, the complexity of the online teacher predictions smoothly increases, and is significantly smaller for most of the epochs. To account for softness differences of the various targets, we consider plotting the adjusted supervision complexity normalized by the target norm $\sqrt{m} \|Y\|_2$. As shown in Fig. 2a, the normalized complexity of offline and online teacher predictions is smaller compared to the dataset labels, indicating a better alignment with top eigenvectors of the LeNet NTK. Importantly, we see that the predictions of an online teacher have significantly lower normalized complexity in the critical early stages of training. Similar observations hold when complexities are measured with respect to a ResNet-20 network (see Appendix D).

**Average teacher complexity.** Ren et al. (2022) observed that teacher predictions fluctuate over time, and showed that using exponentially averaged teachers improves knowledge distillation. Fig. 2c demonstrates that the supervision complexity of an online teacher predictions is always slightly larger than that of the average of predictions of teachers of the last 10 preceding epochs.

**Effect of temperature scaling.** As discussed earlier, higher temperature makes the teacher predictions softer, decreasing their norm. This has a large effect on supervision complexity (Fig. 2b). Even when one controls for the norm of the predictions, the complexity still decreases (Fig. 2b).

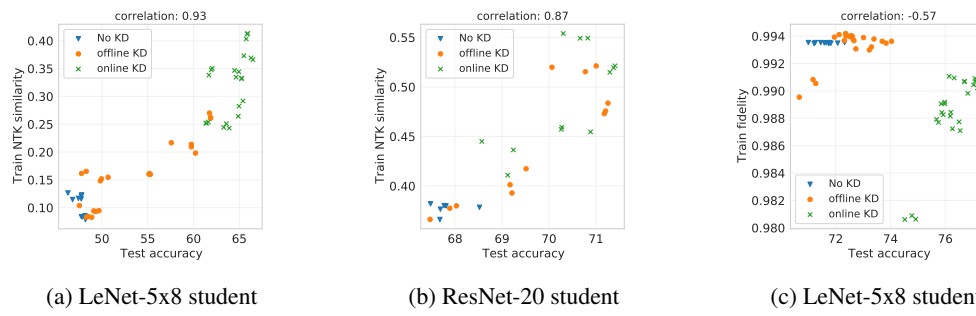

(a) LeNet-5x8 student       (b) ResNet-20 student       (c) LeNet-5x8 student

Figure 3: Relationship between test accuracy, train NTK similarity, and train fidelity for CIFAR-100 students training with either ResNet-56 teacher (panels (a) and (c)) or ResNet-110 (panel (b)).

**NTK similarity.** Remarkably, we observe that across all of our experiments, the final test accuracy of the student is strongly correlated with the similarity of final teacher and student NTKs (see Figures 3, 9, and 10). This cannot be explained by better matching the teacher predictions. In fact, we see that the final fidelity (the rate of classification agreement of a teacher-student pair) measured on training set has no clear relationship with test accuracy. Furthermore, we see that online KD results in better NTK transfer without an explicit regularization loss enforcing such transfer.

## 5 RELATED WORK

Due to space constraints, we briefly discuss the existing works that are the most closely related to our exploration in this paper (see Appendix E for a more comprehensive account of related work.)

**Supervision complexity.** The key quantity in our work is supervision complexity $Y^\top K^{-1} Y$. Cristianini et al. (2001) introduced a related quantity $Y^\top K Y$ called *kernel-target alignment* (KTA) and derived a generalization bound with it for expected Parzen window classifiers. Deshpande et al. (2021) use KTA for model selection in transfer learning. Ortiz-Jiménez et al. (2021) demonstrate that when NTK-target alignment is high, learning is faster and generalizes better. Arora et al. (2019) prove a generalization bound for overparameterized two-layer neural networks with NTK parameterization, trained with gradient flow. Their bound is roughly $\left(Y^\top (K^\infty)^{-1} Y\right)^{1/2} / \sqrt{n}$, where $K^\infty$ is the *expected NTK matrix* at a random initialization. Our bound of Thm. 3 can be seen as a generalization of this result for all kernel methods, including linearized neural networks of any depth and sufficient width, with the only difference of using the empirical NTK matrix. Belkin et al. (2018) warns that bounds based on RKHS complexity of the learned function can fail to explain the good generalization of kernel methods under label noise.

Mobahi et al. (2020) prove that for kernel methods with RKHS norm regularization, self-distillation increases regularization strength. Phuong & Lampert (2019), while studying self-distillation of deep linear networks, derive a bound of the transfer risk that depends on the distribution of the acute angle between teacher parameters and data points. This is in spirit related to supervision complexity as it measures an "alignment" between the distillation objective and data. Ji & Zhu (2020) extend this results to linearized neural networks, showing that $\Delta_z^\top K^{-1} \Delta_z$, where $\Delta_z$ is the logit change during training, plays a key role in estimating the bound. Their bound is qualitatively different than ours, and $\Delta_z^\top K^{-1} \Delta_z$ becomes ill-defined for hard labels.

**Non-static teachers.** Multiple works consider various approaches that train multiple students simultaneously to distill either from each other or from an ensemble (see, e.g., Zhang et al., 2018; Anil et al., 2018; Guo et al., 2020). Zhou et al. (2018) and Shi et al. (2021) train teacher and student together while having a common architecture trunk and regularizing teacher predictions to close that of students, respectively. Jin et al. (2019) study *route constrained optimization* which is closest to the online distillation in §3.2. They employ a few teacher checkpoints to perform a multi-round KD. We complement this line of work by highlighting the role of supervision complexity and by demonstrating that online distillation can be very powerful for students with weak inductive biases.

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

---

**Algorithm 1** Online knowledge distillation.

---

**Require:** Training sample $S$; teacher checkpoints $\{\boldsymbol{g}^{(t_1)}, \ldots, \boldsymbol{g}^{(t_m)}\}$; temperature $\tau > 0$; training steps $T$; minibatch size $b$
1: **for** $t = 1, \ldots, T$ **do**
2:     Draw random $b$-sized minibatch $S'$ from $S$
3:     Compute nearest teacher checkpoint $t^* = \min\{i \in [m] : t_i > t\}$
4:     Update student $\theta \leftarrow \theta - \eta_t \cdot \nabla_\theta \mathcal{L}_{\mathrm{kd-ce}}(f_\theta; \boldsymbol{g}^{(t^*)}, \tau)$ over $S'$
5: **end for**
6: **return** $f_\theta$

---

## A    HYPERPARAMETERS AND IMPLEMENTATION DETAILS

| Dataset | Model | Learning rate |
|---|---|---|
| CIFAR-10, CIFAR-100, binary CIFAR-100 | ResNet-56 (teacher) | 0.1 |
| | ResNet-110 (teacher) | 0.1 |
| | ResNet-20 (CE or MSE students) | 0.1 |
| | LeNet-5x8 (CE or MSE students) | 0.04 |
| Tiny ImageNet | MobileNet-V3-125 (teacher) | 0.04 |
| | ResNet-101 (teacher) | 0.1 |
| | MobileNet-V3-35 (student) | 0.04 |
| | VGG-16 (student) | 0.01 |

Table 4: Initial learning rates for different dataset and model pairs.

In all experiments we use stochastic gradient descent optimizer with 128 batch size and 0.9 Nesterov momentum. The starting learning rates are presented in Table 4. All models for CIFAR datasets are trained for 256 epochs, with a learning schedule that divides the learning rate by 10 at epochs 96, 192, and 224. All models for Tiny ImageNet are trained for 200 epochs, with a learning rate schedule that divides the learning rate by 10 at epochs 75 and 135. The learning rate is warmed-up linearly to its initial value in the first 10 and 5 epochs for CIFAR and Tiny ImageNet models respectively. All VGG and ResNet models use 2e-4 weight decay, while MobileNet models use 1e-5 weight decay.

The LeNet-5 uses ReLU activations. We use the CIFAR variants of ResNets in experiments with CIFAR-10 or (binary) CIFAR-100 datasets. Tiny ImageNet examples are resized to 224x224 resolution to suit the original ResNet, VGG and MobileNet architectures. In all experiments we use standard data augmentation – random cropping and random horizontal flip. In all online learning methods we consider one teacher checkpoint per epoch.

## B    PROOFS

In this appendix we present the deferred proofs. We use the following definition of Rademacher complexity (Mohri et al., 2018).

**Definition 6** (Rademacher complexity). Let $\mathcal{G}$ be a family of functions from $\mathcal{Z}$ to $\mathbb{R}$, and $Z_1, \ldots, Z_n$ be $n$ i.i.d. examples from a distribution $P$ on $\mathcal{Z}$. Then, the *empirical Rademacher complexity* of $\mathcal{G}$ with respect to $(Z_1, \ldots, Z_n)$ is defined as

$$\widehat{\mathfrak{R}}_n(\mathcal{G}) = \mathbb{E}_{\sigma_1, \ldots, \sigma_n} \left[ \sup_{g \in \mathcal{G}} \frac{1}{m} \sum_{i=1}^{n} \sigma_i g(Z_i) \right], \tag{11}$$

where $\sigma_i$ are independent Rademacher random variables (i.e., uniform random variables taking values in $\{-1, 1\}$). The *Rademacher complexity* of $\mathcal{G}$ is then defined as

$$\mathfrak{R}_n(\mathcal{G}) = \mathbb{E}_{Z_1, \ldots, Z_n} \left[ \widehat{\mathfrak{R}}_n(\mathcal{G}) \right]. \tag{12}$$

**Remark 7.** Shifting the hypothesis class $\mathcal{G}$ by a constant function does not change the empirical Rademacher complexity:

$$\widehat{\mathfrak{R}}_n \left(\{f + g : g \in \mathcal{G}\}\right) = \mathbb{E}_{\sigma_1,\ldots,\sigma_n} \left[\sup_{g \in \mathcal{G}} \frac{1}{m} \sum_{i=1}^n \sigma_i \left(f(Z_i) + g(Z_i)\right)\right] \tag{13}$$

$$= \mathbb{E}_{\sigma_1,\ldots,\sigma_n} \left[\frac{1}{m} \sum_{i=1}^n \sigma_i f(Z_i) + \sup_{g \in \mathcal{G}} \frac{1}{m} \sum_{i=1}^n \sigma_i g(Z_i)\right] \tag{14}$$

$$= \mathbb{E}_{\sigma_1,\ldots,\sigma_n} \left[\sup_{g \in \mathcal{G}} \frac{1}{m} \sum_{i=1}^n \sigma_i g(Z_i)\right] = \widehat{\mathfrak{R}}_n(\mathcal{G}). \tag{15}$$

Given the kernel classification setting described in Sec. 2.1, we first prove a slightly more general variant of a classical generalization gap bound in Bartlett & Mendelson (2002, Theorem 21).

**Theorem 8.** *Assume* $\sup_{x \in \mathcal{X}} k(x, x) < \infty$. *Fix any constant* $M > 0$. *Then with probability at least* $1 - \delta$, *every function* $f \in \mathcal{H}$ *with* $\|f\|_{\mathcal{H}} \leq M$ *satisfies*

$$\mathbb{P}_{X,Y}(Y f(X) \leq 0) \leq \frac{1}{n} \sum_{i=1}^n \phi_\gamma(\text{sign}\,(Y_i)\, f(X_i)) + \frac{2M}{\gamma n} \sqrt{\text{Tr}\,(K)} + 3\sqrt{\frac{\ln\,(2/\delta)}{2n}}. \tag{16}$$

*Proof.* Let $\mathcal{F} = \{f \in \mathcal{H} : \|f\| \leq M\}$ and consider the following class of functions:

$$\mathcal{G} = \{(x, y) \mapsto \phi_\gamma(\text{sign}(y)f(x)) : f \in \mathcal{F}\}. \tag{17}$$

By the standard Rademacher complexity classification generalization bound (Mohri et al., 2018, Theorem 3.3), for any $\delta > 0$, with probability at least $1 - \delta$, the following holds for all $f \in \mathcal{F}$:

$$\mathbb{E}_{X,Y} \left[\phi_\gamma(\text{sign}(Y)f(X))\right] \leq \frac{1}{n} \sum_{i=1}^n \phi_\gamma(\text{sign}(Y_i)f(X_i)) + 2\widehat{\mathfrak{R}}_n(\mathcal{G}) + 3\sqrt{\frac{\log(2/\delta)}{2n}}. \tag{18}$$

Therefore, with probability at least $1 - \delta$, for all $f \in \mathcal{F}$

$$\mathbb{P}_{X,Y} \left(Y f(X) \leq 0\right) \leq \frac{1}{n} \sum_{i=1}^n \phi_\gamma(\text{sign}(Y_i)f(X_i)) + 2\widehat{\mathfrak{R}}_n(\mathcal{G}) + 3\sqrt{\frac{\log(2/\delta)}{2n}}. \tag{19}$$

To finish the proof, we upper bound $\widehat{\mathfrak{R}}_n(\mathcal{G})$:

$$\widehat{\mathfrak{R}}_n(\mathcal{G}) = \mathbb{E}_{\sigma_1,\ldots,\sigma_n} \left[\sup_{g \in \mathcal{G}} \frac{1}{n} \sum_{i=1}^n \sigma_i g(X_i, Y_i)\right] \tag{20}$$

$$= \mathbb{E}_{\sigma_1,\ldots,\sigma_n} \left[\sup_{f \in \mathcal{F}} \frac{1}{n} \sum_{i=1}^n \sigma_i \phi_\gamma\left(\text{sign}(Y_i)f(X_i)\right)\right] \tag{21}$$

$$\leq \frac{1}{\gamma} \mathbb{E}_{\sigma_1,\ldots,\sigma_n} \left[\sup_{f \in \mathcal{F}} \frac{1}{n} \sum_{i=1}^n \sigma_i \,\text{sign}(Y_i)f(X_i)\right] \tag{22}$$

$$= \frac{1}{\gamma} \mathbb{E}_{\sigma_1,\ldots,\sigma_n} \left[\sup_{f \in \mathcal{F}} \frac{1}{n} \sum_{i=1}^n \sigma_i f(X_i)\right] \tag{23}$$

$$= \frac{1}{\gamma} \widehat{\mathfrak{R}}_n(\mathcal{F}), \tag{24}$$

where the third line is due to Ledoux & Talagrand (1991). By Lemma 22 of Bartlett & Mendelson (2002), we thus conclude that

$$\widehat{\mathfrak{R}}_n(\mathcal{F}) \leq \frac{M}{n} \sqrt{\text{Tr}\,(K)}. \tag{25}$$

$\square$

### B.1 PROOF OF PROPOSITION 2

*Proof of Proposition 2.* As $K$ is a full rank matrix almost surely, then with probability 1 there exists a vector $\boldsymbol{\alpha} \in \mathbb{R}^n$, such that $K\boldsymbol{\alpha} = \boldsymbol{Y}$. Consider the function $f(x) = \sum_{i=1}^{n} \alpha_i k(X_i, x) \in \mathcal{H}$. Clearly, $f(X_i) = Y_i, \forall i \in [n]$. Furthermore, $\|f\|_{\mathcal{H}}^2 = \boldsymbol{\alpha}^\top K \boldsymbol{\alpha} = \boldsymbol{Y}^\top K^{-1} \boldsymbol{Y}$. The existence of such $f \in \mathcal{H}$ with zero empirical loss and the assumptions on the loss function imply that any optimal solution of problem (3) has a norm at most $\boldsymbol{Y}^\top K^{-1} \boldsymbol{Y}$.[1]  $\square$

### B.2 PROOF OF THM. 3

*Proof.* To get a generalization bound for $f^*$ it is tempting to use Thm. 8 with $M = \|f^*\|$. However, $\|f^*\|$ is a random variable depending on the training data and is an invalid choice for the constant $M$. This issue can be resolved by paying a small logarithmic penalty.

For any $M \geq M_0 = \left\lceil \frac{\gamma \sqrt{n}}{2\sqrt{\kappa}} \right\rceil$ the bound of Thm. 8 is vacuous. Let us consider the set of integers $\mathcal{M} = \{1, 2, \ldots, M_0\}$ and write Thm. 8 for each element of $\mathcal{M}$ with $\delta/M_0$ failure probability. By union bound, we have that with probability at least $1 - \delta$, all instances of Thm. 8 with $M$ chosen from $\mathcal{M}$ hold simultaneously.

If $\boldsymbol{Y}^\top K^{-1} \boldsymbol{Y} \geq M_0$, then the desired bound holds trivially, as the right-hand side becomes at least 1. Otherwise, we set $M = \left\lceil \sqrt{\boldsymbol{Y}^\top K^{-1} \boldsymbol{Y}} \right\rceil \in \mathcal{M}$ and consider the corresponding part of the union bound. We thus have that with at least $1 - \delta$ probability, every function $f \in \mathcal{F}$ with $\|f\| \leq M$ satisfies

$$\mathbb{P}_{X,Y}(Yf(X) \leq 0) \leq \frac{1}{n} \sum_{i=1}^{n} \phi_\gamma(\text{sign}(Y_i) f(X_i)) + \frac{2M}{\gamma n} \sqrt{\text{Tr}(K)} + 3\sqrt{\frac{\ln(2M_0/\delta)}{2n}}.$$

As by Prop. 2 any optimal solution $f^*$ has norm at most $\sqrt{\boldsymbol{Y}^\top K^{-1} \boldsymbol{Y}}$ and $M \leq \sqrt{\boldsymbol{Y}^\top K^{-1} \boldsymbol{Y}} + 1$, we have with probability at least $1 - \delta$,

$$\mathbb{P}_{X,Y}(Yf^*(X) \leq 0) \leq \frac{1}{n} \sum_{i=1}^{n} \phi_\gamma(\text{sign}(Y_i) f^*(X_i)) + \frac{2\sqrt{\boldsymbol{Y}^\top K^{-1} \boldsymbol{Y}} + 2}{\gamma n} \sqrt{\text{Tr}(K)}$$
$$+ 3\sqrt{\frac{\ln(2M_0/\delta)}{2n}}.$$

$\square$

### B.3 PROOF OF PROP. 5

Prop. 5 is a simple corollary of Thm. 3.

*Proof.* We have that
$$P_{X,Y^*}(Y^*f^*(X) \leq 0) = P_{X,Y^*}(Y^*f^*(X) \leq 0 \wedge Y^*g(X) \leq 0)$$
$$+ P_{X,Y^*}(Y^*f^*(X) \leq 0 \wedge Y^*g(X) > 0)$$
$$\leq P_{X,Y^*}(Y^*g(X) \leq 0) + P_X(g(X)f^*(X) \leq 0).$$
The rest follows from bounding $P_X(g(X)f^*(X) \leq 0)$ using Thm. 3.  $\square$

## C  EXTENSION TO MULTICLASS CLASSIFICATION

Let us now consider the case of multiclass classification with $d$ classes. Let $k : \mathcal{X} \times \mathcal{X} \to \mathbb{R}^{d \times d}$ be a matrix-valued positive definite kernel. For every $x \in \mathcal{X}$ and $a \in \mathbb{R}^d$, let $k_x a = k(\cdot, x)a$ be the function from $\mathcal{X}$ to $\mathbb{R}^d$ defined the following way:

$$k_x a(x') = k(x', x)a, \text{ for all } x' \in \mathcal{X}. \tag{26}$$

---

[1]For $[0, 1]$-bounded loss functions, it also holds that $\|f^*\|_{\mathcal{H}}^2 \leq 2/\lambda$. This is not of direct relevance to us, as we will be interested in cases with small $\lambda > 0$.

With any such kernel $k$ there is a unique vector-valued RKHS $\mathcal{H}$ of functions from $\mathcal{X}$ to $\mathbb{R}^d$. This RKHS is the completion of span $\{k_x a : x \in \mathcal{X}, a \in \mathbb{R}^d\}$, with the following inner product:

$$\left\langle \sum_{i=1}^n k_{x_i} a_i, \sum_{j=1}^m k_{x'_j} a'_j \right\rangle_{\mathcal{H}} = \sum_{i=1}^n \sum_{j=1}^m a_i^\top k(x_i, x'_j) a'_j. \tag{27}$$

For any $f \in \mathcal{H}$, the norm $\|f\|_{\mathcal{H}}$ is defined as $\sqrt{\langle f, f \rangle_{\mathcal{H}}}$. Therefore, if $f(x) = \sum_{i=1}^n k_{x_i} a_i$ then

$$\langle f, f \rangle_{\mathcal{H}}^2 = \sum_{i,j=1}^n a_i^\top k(x_i, x_j) a_j \tag{28}$$

$$= \boldsymbol{a}^\top K \boldsymbol{a}, \tag{29}$$

where $\boldsymbol{a}^\top = (a_1^\top, \dots, a_n^\top) \in \mathbb{R}^{nd}$ and

$$K = \begin{bmatrix} k(x_1, x_1) & \cdots & k(x_1, x_n) \\ \cdots & \cdots & \cdots \\ k(x_n, x_1) & \cdots & k(x_n, x_n) \end{bmatrix} \in \mathbb{R}^{nd \times nd}. \tag{30}$$

Suppose $\{(X_i, Y_i)\}_{i \in [n]}$ are $n$ i.i.d. examples sampled from some probability distribution on $\mathcal{X} \times \mathcal{Y}$, where $\mathcal{Y} \subset \mathbb{R}^d$. As in the binary classification case, we consider the regularized kernel problem (3). Let $\boldsymbol{Y}^\top = (Y_1^\top, \dots, Y_n^\top)$ be the concatenation of targets. The following proposition is the analog of Prop. 2 in this vector-valued setting.

**Proposition 9.** *Assume $K$ is full rank almost surely. Assume also $\ell(y, y') \geq 0, \forall y, y' \in \mathcal{Y}$, and $\ell(y, y) = 0, \forall y \in \mathcal{Y}$. Then, with probability 1, for any solution $f^*$ of (3), we have that*

$$\|f^*\|_{\mathcal{H}}^2 \leq \boldsymbol{Y}^\top K^{-1} \boldsymbol{Y}. \tag{31}$$

*Proof.* With probability 1, the kernel matrix $K$ is full rank. Therefore, there exists a vector $\boldsymbol{a}^\top = (a_1^\top, \dots, a_n^\top) \in \mathbb{R}^{nd}$, with $a_i \in \mathbb{R}^d$, such that $K\boldsymbol{a} = \boldsymbol{Y}$. Consider the function $f(x) = \sum_{i=1}^n k_{X_i} a_i \in \mathcal{H}$. Clearly, $f(X_i) = Y_i, \forall i \in [n]$. Furthermore,

$$\|f\|_{\mathcal{H}}^2 = \boldsymbol{a}^\top K \boldsymbol{a} \tag{32}$$

$$= \boldsymbol{Y}^\top K^{-1} \boldsymbol{Y}. \tag{33}$$

The existence of such $f(x) \in \mathcal{H}$ with zero empirical loss and assumptions on the loss function imply that any optimal solution of problem (3) has a norm at most $\boldsymbol{Y}^\top K^{-1} \boldsymbol{Y}$. $\qquad \square$

As in the main text, for a hypothesis $f : \mathcal{X} \to \mathbb{R}^d$ and a labeled example $(x, y)$, let $\rho_f(x, y) = f(x)_y - \max_{y' \neq y} f(x)_{y'}$ be the *prediction margin*. We now restate Thm. 4 and present a proof.

**Theorem 10** (Thm. 4 restated). *Assume that $\kappa = \sup_{x \in \mathcal{X}, y \in [d]} k(x, x)_{y, y} < \infty$, and $K$ is full rank almost surely. Further, assume that $\ell(y, y') \geq 0, \forall y, y' \in \mathcal{Y}$ and $\ell(y, y) = 0, \forall y \in \mathcal{Y}$. Let $M_0 = \lceil \gamma \sqrt{n}/(4d\sqrt{\kappa}) \rceil$. Then, with probability at least $1 - \delta$, for any solution $f^*$ of problem in Eq. (3),*

$$\mathbb{P}_{X,Y}(\rho_{f^*}(X, Y) \leq 0) \leq \frac{1}{n} \sum_{i=1}^n \mathbf{1}\{\rho_{f^*}(X_i, Y_i) \leq \gamma\} + \frac{4d(\boldsymbol{Y}^\top K^{-1} \boldsymbol{Y} + 1)}{\gamma n} \sqrt{\operatorname{Tr}(K)}$$

$$+ 3\sqrt{\frac{\log(2M_0/\delta)}{2n}}. \tag{34}$$

*Proof.* Consider the class of functions $\mathcal{F} = \{f \in \mathcal{H} : \|f\| \leq M\}$ for some $M > 0$. By Theorem 2 of Kuznetsov et al. (2015), for any $\gamma > 0$ and $\delta > 0$, with probability at least $1 - \delta$, the following bound holds for all $f \in \mathcal{F}$:[2]

$$\mathbb{P}_{X,Y}(\rho_f(X, Y) \leq 0) \leq \frac{1}{n} \sum_{i=1}^n \mathbf{1}\{\rho_f(X_i, Y_i) \leq \gamma\} + \frac{4d}{\gamma} \widehat{\mathfrak{R}}_n(\tilde{\mathcal{F}}) + 3\sqrt{\frac{\log(2/\delta)}{2n}}, \tag{35}$$

---

[2]Note that their result is in terms of Rademacher complexity rather than empirical Rademacher complexity. The variant we use can be proved with the same proof, with a single modification of bounding $R(g)$ with empirical Rademacher complexity of $\tilde{\mathcal{G}}$ using Theorem 3.3 of Mohri et al. (2018).

where $\tilde{\mathcal{F}} = \{(x,y) \mapsto f(x)_y : f \in \mathcal{F}, y \in [d]\}$. Next we upper bound the empirical Rademacher complexity of $\tilde{\mathcal{F}}$:

$$\widehat{\mathfrak{R}}_n(\tilde{\mathcal{F}}) = \mathbb{E}_{\sigma_1,\ldots,\sigma_n}\left[\sup_{y\in[d],h\in\mathcal{H},\|h\|\leq M} \frac{1}{n}\sum_{i=1}^{n} \sigma_i h(X_i)_y\right] \tag{36}$$

$$= \mathbb{E}_{\sigma_1,\ldots,\sigma_n}\left[\sup_{y\in[d],h\in\mathcal{H},\|h\|\leq M} \frac{1}{n}\sum_{i=1}^{n} \sigma_i h(X_i)^\top \mathbf{y}\right] \quad (\mathbf{y} \text{ is the one-hot enc. of } y) \tag{37}$$

$$= \mathbb{E}_{\sigma_1,\ldots,\sigma_n}\left[\sup_{y\in[d],h\in\mathcal{H},\|h\|\leq M} \left\langle h, \frac{1}{n}\sum_{i=1}^{n} \sigma_i k_{X_i}\mathbf{y}\right\rangle_{\mathcal{H}}\right] \quad (\text{reproducing property}) \tag{38}$$

$$\leq \frac{M}{n}\mathbb{E}_{\sigma_1,\ldots,\sigma_n}\left[\sup_{y\in[d]} \left\|\sum_{i=1}^{n} \sigma_i k_{X_i}\mathbf{y}\right\|_{\mathcal{H}}\right] \quad (\text{Cauchy-Schwarz}) \tag{39}$$

$$= \frac{M}{n}\sqrt{\mathbb{E}_{\sigma_1,\ldots,\sigma_n}\left[\sup_{y\in[d]} \left\|\sum_{i=1}^{n} \sigma_i k_{X_i}\mathbf{y}\right\|_{\mathcal{H}}^2\right]} \quad (\text{Jensen's inequality}) \tag{40}$$

$$\leq \frac{M}{n}\sqrt{\mathbb{E}_{\sigma_1,\ldots,\sigma_n}\left[\sum_{y=1}^{d} \left\|\sum_{i=1}^{n} \sigma_i k_{X_i}\mathbf{y}\right\|_{\mathcal{H}}^2\right]} \tag{41}$$

$$= \frac{M}{n}\sqrt{\sum_{y=1}^{d}\mathbb{E}_{\sigma_1,\ldots,\sigma_n}\left[\sum_{i=1}^{n} \|\sigma_i k_{X_i}\mathbf{y}\|_{\mathcal{H}}^2 + \sum_{i\neq j}\langle \sigma_i k_{X_i}\mathbf{y}, \sigma_j k_{X_j}\mathbf{y}\rangle\right]} \tag{42}$$

$$= \frac{M}{n}\sqrt{\sum_{y=1}^{d}\mathbb{E}_{\sigma_1,\ldots,\sigma_n}\left[\sum_{i=1}^{n} \|\sigma_i k_{X_i}\mathbf{y}\|_{\mathcal{H}}^2\right]} \quad (\text{independence of } \sigma_i) \tag{43}$$

$$= \frac{M}{n}\sqrt{\sum_{y=1}^{d}\left[\sum_{i=1}^{n} \mathbf{y}^\top k(X_i, X_i)\mathbf{y}\right]} \tag{44}$$

$$= \frac{M}{n}\sqrt{\mathrm{Tr}(K)}. \tag{45}$$

The proof is concluded with the same reasoning of the proof of Thm. 3. □

## D  ADDITIONAL RESULTS AND DISCUSSION

In this appendix we present additional results and discussion to support the main findings of this work.

**On early stopped teachers.** Cho & Hariharan (2019) observe that sometimes offline KD works better with early stopped teachers. Such teachers have worse accuracy and perhaps results in a smaller student margin, but they also have a significantly smaller supervision complexity (see Fig. 2a), which provides a possible explanation for this phenomenon.

**Teaching students with weak inductive biases.** As we saw earlier, a fully trained teacher can have predictions as complex as random labels for a weak student at initialization. This low alignment of student NTK and teacher predictions can result in memorization. In contrast, an early stopped teacher captures simple patterns and has a better alignment with the student NTK, allowing the student to learn these patterns in a generalizable fashion. This feature learning improves the student NTK and allows learning more complex patterns in future iterations. We hypothesize that this is the mechanism that allows online distillation to outperform offline distillation in some cases.

Table 5: Results on CIFAR-10. Every second line is an MSE student.

| Setting | No KD | Offline KD | | Online KD | | Teacher |
|---|---|---|---|---|---|---|
| | | $\tau = 1$ | $\tau = 4$ | $\tau = 1$ | $\tau = 4$ | |
| ResNet-56 $\to$ LeNet-5x8 | $81.8 \pm 0.5$ | $82.4 \pm 0.5$ | $86.0 \pm 0.2$ | $86.8 \pm 0.2$ | $88.6 \pm 0.1$ | 93.2 |
| ResNet-56 $\to$ LeNet-5x8 | $83.4 \pm 0.3$ | $83.1 \pm 0.2$ | $84.9 \pm 0.1$ | $85.6 \pm 0.1$ | $87.1 \pm 0.1$ | 93.2 |
| ResNet-110 $\to$ LeNet-5x8 | $81.7 \pm 0.3$ | $81.9 \pm 0.5$ | $85.8 \pm 0.1$ | $86.5 \pm 0.1$ | $88.8 \pm 0.1$ | 93.9 |
| ResNet-110 $\to$ LeNet-5x8 | $83.2 \pm 0.4$ | $83.2 \pm 0.1$ | $85.0 \pm 0.3$ | $85.6 \pm 0.1$ | $87.1 \pm 0.2$ | 93.9 |
| ResNet-110 $\to$ ResNet-20 | $91.4 \pm 0.2$ | $91.4 \pm 0.1$ | $92.8 \pm 0.0$ | $92.2 \pm 0.3$ | $93.1 \pm 0.1$ | 93.9 |
| ResNet-110 $\to$ ResNet-20 | $90.9 \pm 0.1$ | $90.9 \pm 0.2$ | $91.6 \pm 0.2$ | $91.2 \pm 0.1$ | $92.1 \pm 0.2$ | 93.9 |

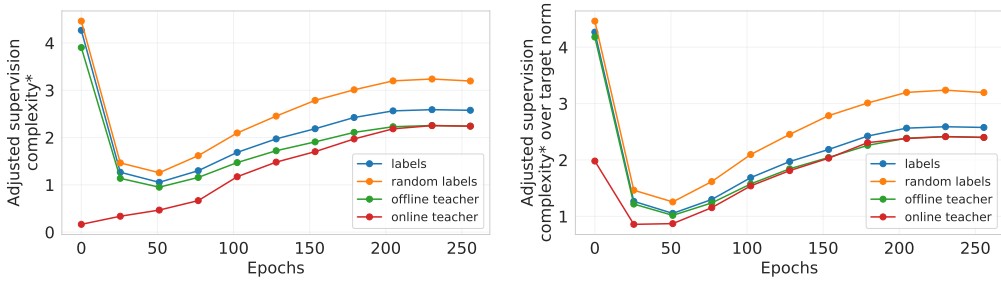

Figure 4: Adjusted supervision complexities* of various targets with respect to a LeNet-5x8 network at different stages of its training. The experimental setup of the left and right plots matches that of Fig. 1c and Fig. 2a respectively.

**CIFAR-10 results.** Table 5 presents the comparison of standard training, offline distillation, and online distillation on CIFAR-10. We see that the results are qualitatively similar to CIFAR-100 results.

**Comparison of supervision complexities.** In the main text, we have introduced the notion of *adjusted supervision complexity*, which for a given neural network $f(x)$ with NTK $k$ and a set of labeled examples $\{(X_i, Y_i)\}_{i \in [m]}$ is defined as:

$$\frac{1}{n}\sqrt{(\boldsymbol{Y} - f(X_{1:m}))^\top K^{-1}(\boldsymbol{Y} - f(X_{1:m})) \cdot \operatorname{Tr}(K)}. \tag{46}$$

As discussed earlier, the subtraction of initial predictions is the appropriate way to measure complexity given the form of the optimization problem (9). Nevertheless, it is meaningful to consider the following quantity as well:

$$\frac{1}{n}\sqrt{\boldsymbol{Y}^\top K^{-1}\boldsymbol{Y} \cdot \operatorname{Tr}(K)}, \tag{47}$$

in order to measure "alignment" of targets $\boldsymbol{Y}$ with the NTK $k$. We call this quantity *adjusted supervision complexity\**. We compare the adjusted supervision complexities* of random labels, dataset labels, and predictions of an offline and online ResNet-56 teacher predictions with respect to various checkpoints of the LeNet-5x8 network. The results presented in Fig. 4 are remarkably similar to the results with adjusted supervision complexity (Fig. 1c and Fig. 2a). We therefore, focus only on adjusted supervision complexity of (46) when comparing various targets. The only other experiment where we compute adjusted supervision complexities* (i.e., without subtracting the current predictions from labels) is presented in Fig. 7, where the goal is to demonstrate that training labels become aligned with the training NTK matrix over the course of training.

Fig. 5 presents the comparison of adjusted supervision complexities, but with respect to a ResNet-20 network, instead of a LeNet-5x8 network. Again, we see that in the early epochs dataset labels and offline teacher predictions are almost as complex as random labels. Unlike the case of the LeNet-5x8 network, random labels, dataset labels, and offline teacher predictions do not exhibit a U-shaped behavior. As for the LeNet-5x8 network, the shape of these curves is in agreement with the behavior of the condition number of the NTK (Fig. 6). Importantly, we still observe that the complexity of

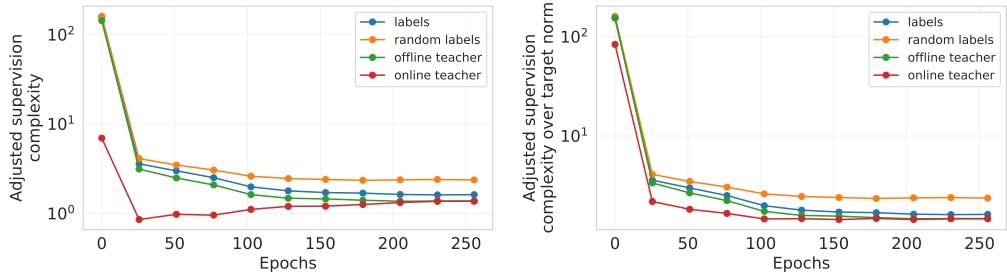

Figure 5: Adjusted supervision complexities of various targets with respect to a *ResNet-20* network at different stages of its training. Besides the network choice, the experimental setup of the left and right plots matches that of Fig. 1c and Fig. 2a respectively. Note that the y-axes are in logarithmic scale.

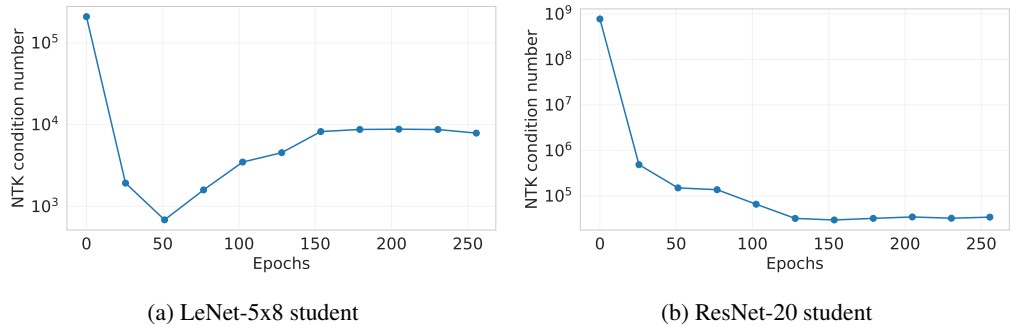

(a) LeNet-5x8 student    (b) ResNet-20 student

Figure 6: Condition number of the NTK matrix of a LeNet5x8 (a) and ResNet-20 (b) students trained with MSE loss on binary CIFAR-100. The NTK matrices are computed on $2^{12}$ test examples.

the online teacher predictions is significantly smaller compared to the other targets, even when we account for the norm of predictions.

**The effect of frequency of teacher checkpoints.**   As mentioned earlier, throughout this paper we used one teacher checkpoint per epoch. While this served our goal of establishing efficacy of online distillation, this choice is prohibitive for large teacher networks. To understand the effect of the frequency of teacher checkpoints, we conduct an experiment on CIFAR-100 with ResNet-56 and LeNet-5x8 student with varying frequency of teacher checkpoints. In particular, we consider checkpointing the teacher once in every $\{1, 2, 4, 8, 16, 32, 64, 128\}$ epochs. The results presented in Fig. 8 show that reducing the teacher checkpointing frequency to once in 16 epochs results in only a minor performance drop for online distillation with $\tau = 4$.

**On label supervision in KD.**   So far in all distillation methods dataset labels were not used as an additional source of supervision for students. However, in practice it is common to train a student with a convex combination of knowledge distillation and standard losses: $(1 - \alpha)\mathcal{L}_{\mathrm{ce}} + \alpha\mathcal{L}_{\mathrm{kd\text{-}ce}}$. To verify that the choice of $\alpha = 1$ does not produce unique conclusions regarding efficacy of online distillation, we do experiments on CIFAR-100 with varying values of $\alpha$. The results presented in Table 6 confirm our main conclusions on online distillation. Furthermore, we observe that picking $\alpha = 1$ does not result in significant degradation of student performance.

**The effect of $\tau$.**   To confirm that our main conclusions regarding online distillation do not depend on the temperature value, we present additional experiments on CIFAR-100 with $\tau = 2$ in Table 7.

**NTK similarity.**   Figures 9 and 10 presents additional evidence that (a) training NTK similarity of the final student and the teacher is correlated with the final student test accuracy; and (b) that online distillation manages to transfer teacher NTK better.

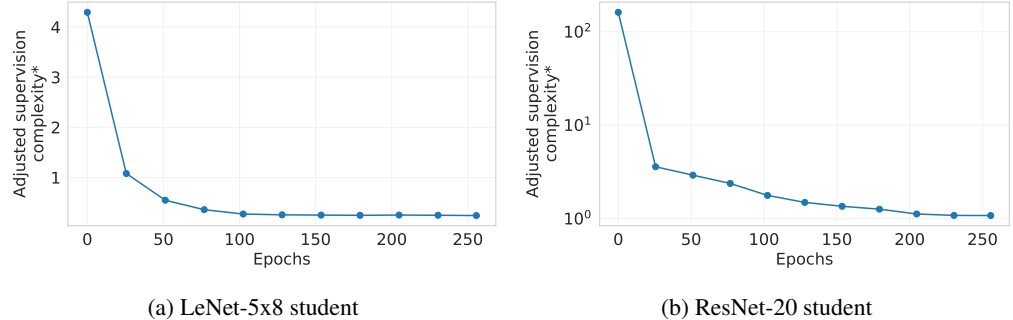

(a) LeNet-5x8 student          (b) ResNet-20 student

Figure 7: Adjusted supervision complexity* of dataset labels measured on a subset of $2^{12}$ *training* examples of binary CIFAR-100. Complexities are measured with respect to either a LeNet-5x8 (on the left) or ResNet-20 (on the right) models trained with MSE loss and without knowledge distillation. Note that the plot on the right is in logarithmic scale.

| Teacher update period in epochs | Online KD $\tau = 1$ | $\tau = 4$ |
|---|---|---|
| 1 (the default value) | $61.9 \pm 0.2$ | $66.1 \pm 0.4$ |
| 2 | $61.5 \pm 0.4$ | $66.0 \pm 0.3$ |
| 4 | $61.4 \pm 0.2$ | $65.6 \pm 0.2$ |
| 8 | $60.0 \pm 0.3$ | $65.4 \pm 0.0$ |
| 16 | $59.3 \pm 0.6$ | $65.4 \pm 0.0$ |
| 32 | $56.9 \pm 0.0$ | $64.1 \pm 0.4$ |
| 64 | $55.5 \pm 0.4$ | $62.8 \pm 0.7$ |
| 128 | $51.4 \pm 0.5$ | $61.3 \pm 0.1$ |

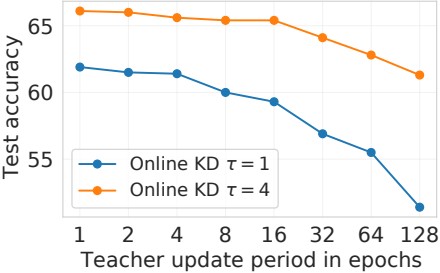

Figure 8: Online KD results for a LeNet-5x8 student on CIFAR-100 with varying frequency of a ResNet-56 teacher checkpoints.

Table 6: Knowledge distillation results on CIFAR-100 with varying loss mixture coefficient $\alpha$.

| Setting | $\alpha$ | No KD | Offline KD $\tau = 1$ | $\tau = 4$ | Online KD $\tau = 1$ | $\tau = 4$ | Teacher |
|---|---|---|---|---|---|---|---|
| ResNet-56 → LeNet-5x8 | 0.2 | | $47.6 \pm 0.7$ | $57.6 \pm 0.2$ | $54.3 \pm 0.7$ | $59.0 \pm 0.6$ | |
| | 0.4 | | $48.9 \pm 0.3$ | $58.9 \pm 0.4$ | $56.7 \pm 0.5$ | $62.5 \pm 0.2$ | |
| | 0.6 | $47.3 \pm 0.6$ | $49.4 \pm 0.5$ | $59.7 \pm 0.0$ | $61.1 \pm 0.0$ | $65.3 \pm 0.2$ | 72.0 |
| | 0.8 | | $49.8 \pm 0.1$ | $60.1 \pm 0.1$ | $62.0 \pm 0.1$ | $65.9 \pm 0.2$ | |
| | 1.0 | | $50.1 \pm 0.4$ | $59.9 \pm 0.2$ | $61.9 \pm 0.2$ | $66.1 \pm 0.4$ | |
| ResNet-56 → ResNet-20 | 0.2 | | $67.9 \pm 0.3$ | $70.3 \pm 0.3$ | $68.2 \pm 0.3$ | $70.3 \pm 0.1$ | |
| | 0.4 | | $67.9 \pm 0.1$ | $71.0 \pm 0.2$ | $68.7 \pm 0.2$ | $71.4 \pm 0.2$ | |
| | 0.6 | $67.7 \pm 0.5$ | $68.1 \pm 0.3$ | $71.3 \pm 0.1$ | $69.6 \pm 0.4$ | $71.5 \pm 0.2$ | 72.0 |
| | 0.8 | | $68.3 \pm 0.2$ | $71.4 \pm 0.4$ | $69.8 \pm 0.3$ | $71.1 \pm 0.3$ | |
| | 1.0 | | $68.2 \pm 0.3$ | $71.6 \pm 0.2$ | $69.6 \pm 0.3$ | $71.4 \pm 0.3$ | |

Table 7: Results on CIFAR-100.

| Setting | No KD | Offline KD $\tau = 1$ | $\tau = 2$ | Online KD $\tau = 1$ | $\tau = 2$ | Teacher |
|---|---|---|---|---|---|---|
| ResNet-56 → LeNet-5x8 | $47.3 \pm 0.6$ | $50.1 \pm 0.4$ | $55.2 \pm 0.1$ | $61.9 \pm 0.2$ | $64.7 \pm 0.2$ | 72.0 |
| ResNet-56 → ResNet-20 | $67.7 \pm 0.5$ | $68.2 \pm 0.3$ | $70.4 \pm 0.3$ | $69.6 \pm 0.3$ | $70.8 \pm 0.3$ | 72.0 |
| ResNet-110 → LeNet-5x8 | $47.2 \pm 0.5$ | $48.6 \pm 0.8$ | $54.0 \pm 0.5$ | $60.8 \pm 0.2$ | $63.9 \pm 0.2$ | 73.4 |
| ResNet-110 → ResNet-20 | $67.8 \pm 0.3$ | $67.8 \pm 0.2$ | $69.3 \pm 0.2$ | $69.0 \pm 0.3$ | $70.5 \pm 0.3$ | 73.4 |

## E  EXPANDED DISCUSSION ON RELATED WORK

The key contributions of this work are the demonstration of the role of supervision complexity in student generalization, and the establishment of online knowledge distillation as a theoretically grounded and effective method. Both supervision complexity and online distillation have a number of relevant precedents in the literature that are worth comment.

*Transferring knowledge beyond logits*. In the seminal works of Buciluǎ et al. (2006); Hinton et al. (2015) transferred "knowledge" is in the form of output probabilities. Later works suggest other notions of "knowledge" and other ways of transferring knowledge (Gou et al., 2021). These include activations of intermediate layers (Romero et al., 2015), attention maps (Zagoruyko & Komodakis, 2017), classifier head parameters (Chen et al., 2022), and various notions of example similarity (Passalis & Tefas, 2018; Park et al., 2019; Tung & Mori, 2019; Tian et al., 2020; He & Ozay, 2021). Transferring teacher NTK matrix belongs to this latter category of methods. Zhang et al. (2022) propose to transfer a low-rank approximation of a feature map corresponding the teacher NTK.

*Non-static teachers*. Some works on KD consider non-static teachers. In order to bridge teacher-student capacity gap, Mirzadeh et al. (2020) propose to perform a few rounds of distillation with teachers of increasing capacity. In deep mutual learning (Zhang et al., 2018; Chen et al., 2020), codistillation (Anil et al., 2018), and collaborative learning (Guo et al., 2020), multiple students are trained simultaneously, distilling from each other or from an ensemble. In Zhou et al. (2018) and Shi et al. (2021), the teacher and the student are trained together. In the former they have a common architecture trunk, while in the latter the teacher is penalized to keep its predictions close to the student's predictions. The closest method to the online distillation method of this work is *route constrained optimization* (Jin et al., 2019), where a few teacher checkpoints are selected for a multi-round distillation. Rezagholizadeh et al. (2022) employ a similar procedure but with an annealed temperature that decreases linearly with training time, followed by a phase of training with dataset labels only. The idea of distilling from checkpoints also appears in Yang et al. (2019), where a network is trained with a cosine learning rate schedule, simultaneously distilling from the checkpoint of the previous learning rate cycle.

*Fundamental understanding of distillation*. The effects of temperature, teacher-student capacity gap, optimization time, data augmentations, and other training details is non-trivial (Cho & Hariharan, 2019; Beyer et al., 2022; Stanton et al., 2021). It has been hypothesized and shown to some extent that teacher soft predictions capture class similarities, which is beneficial for the student (Hinton et al., 2015; Furlanello et al., 2018; Tang et al., 2020). Yuan et al. (2020) demonstrate that this softness of teacher predictions also has a regularization effect, similar to label smoothing. Menon et al. (2021) argue that teacher predictions are sometimes closer to the Bayes classifier than the hard labels of the dataset, reducing the variance of the training objective. The vanilla knowledge distillation loss also introduces some optimization biases. Mobahi et al. (2020) prove that for kernel methods with RKHS norm regularization, self-distillation increases regularization strength, resulting in smaller norm RKHS norm solutions.

Phuong & Lampert (2019) prove that in a self-distillation setting, deep linear networks trained with gradient flow converge to the projection of teacher parameters into the data span, effectively recovering teacher parameters when the number of training points is large than the number of parameters. They derive a bound of the transfer risk that depends on the distribution of the acute angle between teacher parameters and data points. This is in spirit related to supervision complexity as it measures an "alignment " between the distillation objective and data Ji & Zhu (2020) extend this results to linearized neural networks, showing that the quantity $\Delta_z^\top K^{-1} \Delta_z$, where $\Delta_z$ is the logit change during training, plays a key role in estimating the bound. The resulting bound is qualitatively different compared to ours, and the $\Delta_z^\top K^{-1} \Delta_z$ becomes ill-defined for hard labels.

*Supervision complexity*. The key quantity in our work is $Y^\top K^{-1} Y$. Cristianini et al. (2001) introduced a related quantity $Y^\top K Y$ called *kernel-target alignment* and derived a generalization bound with it for expected Parzen window classifiers. As an easy-to-compute proxy to supervision complexity, Deshpande et al. (2021) use kernel-target alignment for model selection in transfer learning. Ortiz-Jiménez et al. (2021) demonstrate that when NTK-target alignment is high, learning is faster and generalizes better. Arora et al. (2019) prove a generalization bound for overparameterized two-layer neural networks with NTK parameterization trained with gradient flow. Their bound

is approximately $\sqrt{\boldsymbol{Y}^\top (K^\infty)^{-1} \boldsymbol{Y}}/\sqrt{n}$, where $K^\infty$ is the *expected NTK matrix* at a random initialization. Our bound of Thm. 3 can be seen as a generalization of this result for all kernel methods, including linearized neural networks of any depth and sufficient width, with the only difference of using the empirical NTK matrix. Belkin et al. (2018) warns that bounds based on RKHS complexity of the learned function can fail to explain the good generalization capabilities of kernel methods in presence of label noise.

*Future work.* There are several potential directions for future work. *Adaptive temperature* scaling for online distillation, where the teacher predictions are smoothened so as to ensure low target complexity, is one such direction. Another avenue is to explore alternative ways to smoothen teacher prediction besides temperature scaling; e.g., can one perform *sample-dependent* scaling? There is large potential for improving online KD by making more informed choices for the frequency and positions of teacher checkpoints, and controlling how much the student is trained in between teacher updates. Finally, while we demonstrated that online distillation results in a better alignment with the teacher's NTK matrix, understanding *why* this happens is an open and interesting problem.

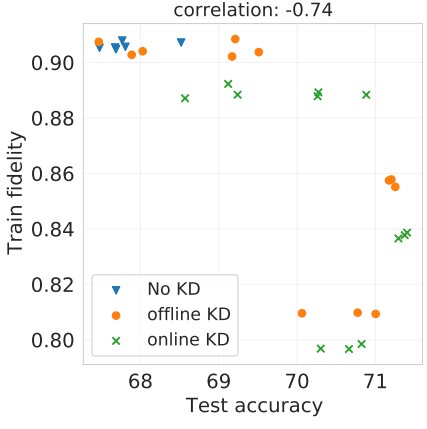

(a) CIFAR-100, ResNet-20 student, ResNet-110 teacher

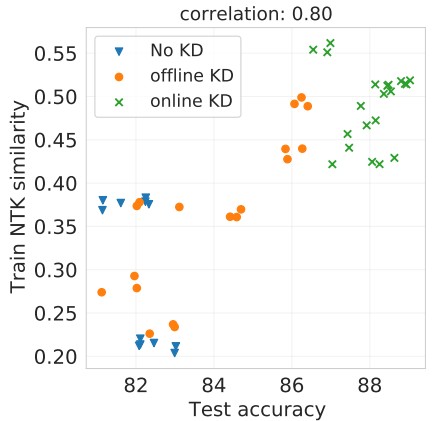

(b) CIFAR-10, LeNet-5x8 student, ResNet-56 teacher

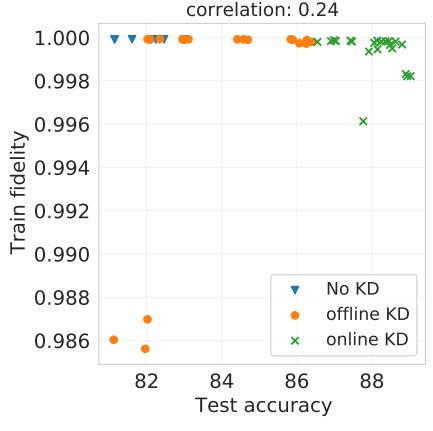

(c) CIFAR-10, LeNet-5x8 student, ResNet-56 teacher

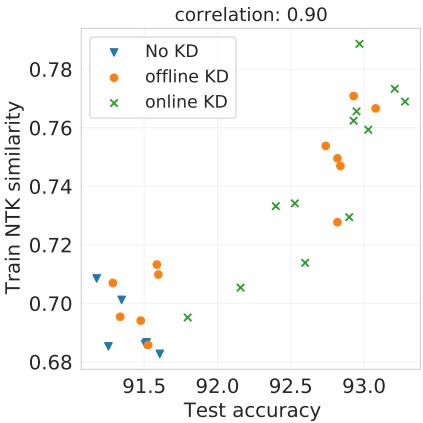

(d) CIFAR-10, ResNet-20 student, ResNet-101 teacher

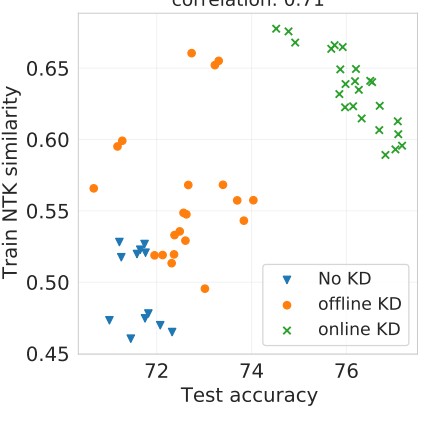

(e) Binary CIFAR-100, LeNet-5x8 student, ResNet-56 teacher

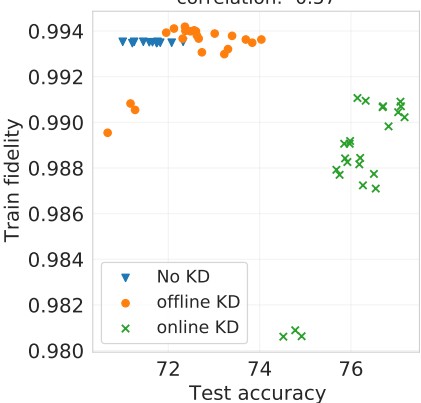

(f) Binary CIFAR-100, LeNet-5x8 student, ResNet-56 teacher

Figure 9: Relationship between test accuracy, train NTK similarity, and train fidelity for various teacher, student, and dataset configurations.

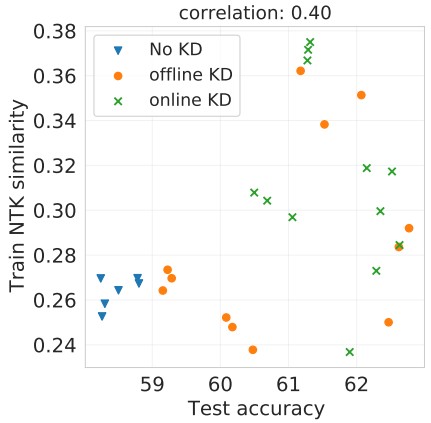

(a) Tiny ImageNet, MobileNet-V3-35 student, MobileNet-V3-125 teacher

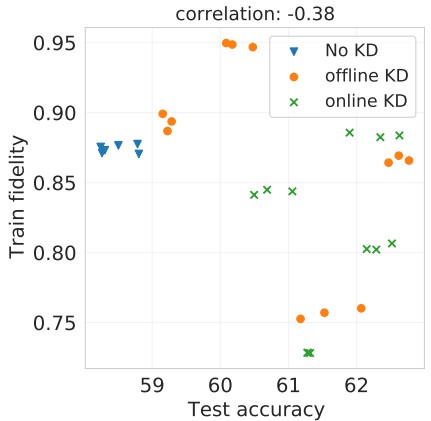

(b) Tiny ImageNet, MobileNet-V3-35 student, MobileNet-V3-125 teacher

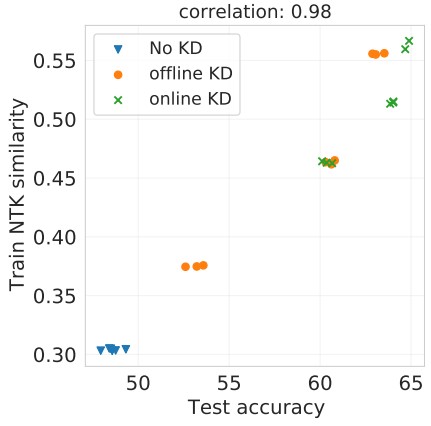

(c) Tiny ImageNet, VGG-16 student, ResNet-101 teacher

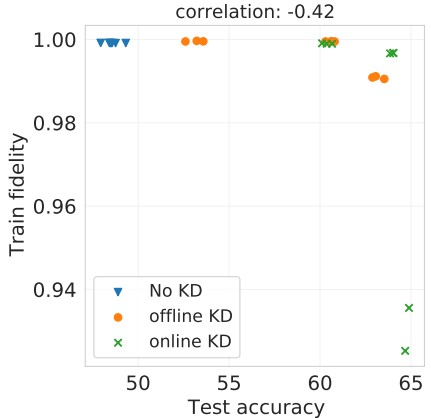

(d) Tiny ImageNet, VGG-16 student, ResNet-101 teacher

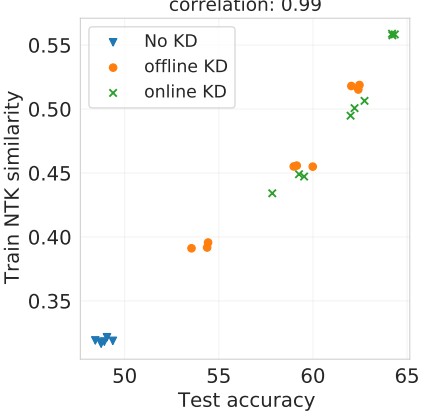

(e) Tiny ImageNet, VGG-16 student, MobileNet-V3-125 teacher

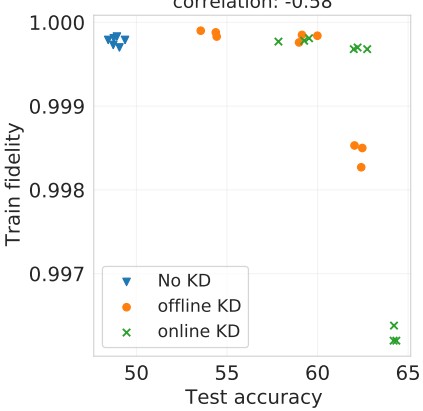

(f) Tiny ImageNet, VGG-16 student, MobileNet-V3-125 teacher

Figure 10: Relationship between test accuracy, train NTK similarity, and train fidelity for various teacher, student, and dataset configurations.

