# OpenReview forum: "Supervision Complexity and its Role in Knowledge Distillation"
_ICLR.cc/2023/Conference — ICLR 2023 poster_

### Official Review · Reviewer_AC7X · 2022-10-22

**Confidence:** 2
**Correctness:** 4
**Technical Novelty And Significance:** 3
**Empirical Novelty And Significance:** 2
**Recommendation:** 8

**Clarity, Quality, Novelty And Reproducibility:**

As mentioned above, the theoretical analysis section is dense without examples or discussions on the intuitive ideas of the proposal. This part is hard to follow.

**Strength And Weaknesses:**

Strengths:
1. The paper offers new theoretical insights on the learning process of KD including the role of temperature scaling and early-stopping, how to teach a weak student, and the effectiveness of online KD.

2. An online KD algorithm is proposed based on the analysis which is shown to be effective on real datasets.

Weakness:
1. The theoretical analysis section is dense without examples or discussions on the intuitive ideas of the proposal.

2. The experimental results are not very strong. The proposed technique shows marginal improvements on many cases in the overall result tables (Tables 1 to 3).

**Summary Of The Paper:**

This paper proposes a knowledge distillation (KD) framework that leverages supervision complexity to measure the alignment between teacher-provided supervision and the student’s neural tangent kernel. It first shows how supervision complexity is defined and measured for kernel-based classifiers, which is then extended to neural classifiers. Then, an online KD algorithm is proposed based on the analysis. Experimental results on real datasets show that the proposed techniques are effective.

**Summary Of The Review:**

The paper makes theoretical contributions by offering new insights into the learning process of KD including the role of temperature scaling and early-stopping, how to teach a weak student, and the effectiveness of online KD. It has the potential to lead to new studies with improved KD learning process and outcomes.

=== Update after rebuttal ==

Thank you for the response. There are no further questions.

---

> ### Author Response · Authors · 2022-11-11
> **Response to Reviewer AC7X**
>
> We would like to thank the reviewer for useful comments and feedback.
>
> > 1. The theoretical analysis section is dense without examples or discussions on the intuitive ideas of the proposal.
>
> We have added more details in the “Implications for neural classifiers” part of Section 2.2. We will continue thinking how to improve Section 2 further.
>
> > 2. The experimental results are not very strong. The proposed technique shows marginal improvements on many cases in the overall result tables (Tables 1 to 3).
>
> Please see the common response.

---

### Official Review · Reviewer_oX8Z · 2022-10-25

**Confidence:** 4
**Correctness:** 3
**Technical Novelty And Significance:** 3
**Empirical Novelty And Significance:** 3
**Recommendation:** 5

**Clarity, Quality, Novelty And Reproducibility:**

Paper is easy to follow and has some interesting observations w.r.t. student generalization behavior during knowledge distiilation.



**Strength And Weaknesses:**


Strengths:
- Generalization bound (Prop.5) on the student risk in terms of three quantities : (a) teacher  risk, (b) students empirical margin w.r.t. teacher predictions, and (c) complexity of the teacher predictions.
- Proposed notion of supervision complexity $Y^\top K^{-1} Y$, up to a certain extent, explains the KD behavior w.r.t. temperature scaling and early stopping

Weaknesses:
- Online KD is storage wise expensive as it stores intermediate teacher checkpoints
- Empirically, at least in moderate capacity gap between teacher-student, it does not seem to be achieving much gains ( see Table~1, student=ResNet-20 )


Questions for Authors:

(1) Why do some of the experiments use temperatue = 1 and 2.. while others use temperature=1 and 4?

(2) Online KD seems to store all checkpoints of a teacher. This is infeasible in many real-world applications. Typically you will have access to only the final few teacher checkpoints. How do you explain this overhead and its practicality?

(3) Appendix notes that you consider one teacher checkpoint per epoch. This does not seem very practical. Have you tried any ablations where relatively fewer teacher checkpoints were used. Alternatively, you could train a teacher in parallel to the student (although it is much more computationally expensive to train a large teacher), and avoid the storage overhead.

(4) In Table.3, MSE loss seems to outperform the CE loss. Why is the same loss not used in the Table.1 or 2? Let me know if this exists somewhere in the supplementary.

(5) Why are you using different teachers and students in CIFAR-100, Tiny-Imagenet? why not use the widely used teacher-student combinations?

(6) What is the value of alpha in the KD formulation? This hyper-parameter has not been mentioned anywhere that trades off the CE and KL-divergence loss.

(7) How do you explain the gains in binary classification CIFAR100 experiments, if not for the so called "dark knowledge"?

(8) Prop.5 explains the effect of temperature on the KD generalization. How does this explain the effect of early stopping? Similarly, how does this explain the effect of teaching a weak student?

(9) Why doesn't the NTK result by Arora et al (2019) directly give you the supervision complexity term? It has the form $Y^\top (K^{\inf})^{-1} Y$.


(10) Why are you using different teachers and students in CIFAR-100, Tiny-Imagenet? why not use the widely used teacher-student combinations (see Sim-KD (Knowledge Distillation with the Reused Teacher Classifier https://github.com/DefangChen/SimKD) and references therein )?


**Summary Of The Paper:**

This work studies the generalization characteristics of Knowledge-Distillation (KD) by proposing a new theoretical framework that leverages supervision complexity: a measure of the alignment between teacher supervision and the student's neural tangent kernel. It provides a justification for the temperature scaling and early stopping often used in the KD literature for superior performance. Further, it proposes Online Distillation, where at each epoch i in training, student uses the teacher model obtained at epoch i. Thus it progressively increases the complexity of teacher supervision, where the final epoch teacher is the supervision used in the vanilla KD in all training epochs. Finally, the online KD procedure and the theoretical findings are validated on benchmark datasets.

**Summary Of The Review:**

This paper introduces  supervision complexity to help explain student generalization behavior during knowledge distiilation. It provides insights into temperature scaling and early stopping, used frequently to obtain better performing student models. Novelty of the supervision complexity is somewhat lacking as there have been attempts to explain similar behavior (NTK literature has a similar term). Main bottleneck in the proposed online KD scheme is the expensive storage of intermediate teacher checkpoints which may become prohibitive once you go beyond the toy setups of CIFAR/Tiny-Imagenet problems. In addition, even the distillation gains shown for these datasets are not that significant.

---

> ### Author Response · Authors · 2022-11-11
> **Response to Reviewer oX8Z (Part I)**
>
> We would like to thank the reviewer for the thorough reading of our work.
>
> > Weakness 1. Online KD is storage wise expensive as it stores intermediate teacher checkpoints.
> > (1) Online KD seems to store all checkpoints of a teacher. This is infeasible in many real-world applications. Typically you will have access to only the final few teacher checkpoints. How do you explain this overhead and its practicality?
> > (3) Appendix notes that you consider one teacher checkpoint per epoch. This does not seem very practical. Have you tried any ablations where relatively fewer teacher checkpoints were used. Alternatively, you could train a teacher in parallel to the student (although it is much more computationally expensive to train a large teacher), and avoid the storage overhead.
>
> We wish to first emphasize that similar algorithms to the online KD in this paper have been considered in several prior works (Zhou et al. (2018), Shi et al. (2021), Jin et al. (2019), Rezagholizadeh et al. (2022)).
>
> Indeed, the version of online distillation we consider is expensive in terms of memory, as one needs to store many teacher checkpoints. This was an intentional choice to support our theory that increasing supervision complexity smoothly can be beneficial for student performance. The existing works on online distillation try to make online distillation memory efficient by picking only a few teacher checkpoints or doing the distillation while training the teacher. We didn’t pursue these directions in order to not deviate from our main goal of highlighting the role of supervision complexity in knowledge distillation. Nevertheless, we had done preliminary experiments to understand how the frequency of checkpointing affects student performance. We have included these results in the updated manuscript (Figure 8). In the particular case of online distillation on CIFAR-100 with ResNet-56 teacher and LeNet-5x8 student, we see that reducing the teacher checkpointing frequency to once in 16 epochs results in only a minor performance drop for $\\tau=4$.
>
>
> > Weakness 2. Empirically, at least in moderate capacity gap between teacher-student, it does not seem to be achieving much gains ( see Table~1, student=ResNet-20 ).
>
> Please see the common response.
>
>
> > (1) Why do some of the experiments use temperature = 1 and 2.. while others use temperature=1 and 4?
>
> In our preliminary experiments, we tried 4 values for temperature: 1, 2, 4, and 8. The latter was too large for all datasets. In all cases besides Tiny-ImageNet, $\\tau=4$ was the optimal choice for both online KD and offline KD. We noticed that the larger the number of classes is, the worse high temperature values start to perform. In particular, for Tiny-ImageNet (200 classes) the optimal value was $\\tau=2$.
>
> To confirm that our main conclusions regarding online distillation do not depend on the temperature value, we did additional experiments on CIFAR-100 with $\\tau=2$. The results are presented in Table 7 of the updated manuscript.
>
>
> > (4) In Table.3, MSE loss seems to outperform the CE loss. Why is the same loss not used in the Table.1 or 2? Let me know if this exists somewhere in the supplementary.
>
> In practice it is common to use the CE loss. We consider the MSE loss too as it suits our theoretical observations better by allowing for interpolation (i.e., the training loss can be made 0). The MSE loss is considered only in those cases for which we have supervision complexity plots. In Tables 1 and 2 the main goal is to demonstrate that online KD can outperform offline distillation significantly in some cases.
>
>
> > (5) Why are you using different teachers and students in CIFAR-100, Tiny-Imagenet? why not use the widely used teacher-student combinations?
> > (10) Why are you using different teachers and students in CIFAR-100, Tiny-Imagenet? why not use the widely used teacher-student combinations (see Sim-KD (Knowledge Distillation with the Reused Teacher Classifier https://github.com/DefangChen/SimKD) and references therein )?
>
> The teacher and student choices for CIFAR-100 and Tiny-ImageNet are different from each other as Tiny-ImageNet is a harder dataset. In general, our goal in selection of teacher-student pairs was to have cases with large teacher-student capacity gaps or architectural differences. Had we considered more standard configurations, we would miss interesting cases such as the LeNet-5x8 student for CIFAR-100 and the VGG-16 student for Tiny-ImageNet, where the student has much weaker inductive biases and the effect of online distillation is large (14-19 percentage point test accuracy increase compared to standard training).

---

> > ### Author Response · Authors · 2022-11-11
> > **Response to Reviewer oX8Z (Part II)**
> >
> > > (6) What is the value of alpha in the KD formulation? This hyper-parameter has not been mentioned anywhere that trades off the CE and KL-divergence loss.
> >
> > In all knowledge distillation experiments reported in this work, we don't use dataset labels as an additional source of supervision. We noted this following Equation 2 of the paper, but have now made this more explicit in the experiments section.
> >
> > We did not use dataset labels to minimize differences between the theory and experiments. Such a choice has also been made in prior theoretical analyses, e.g., (Menon et al., 2021) and (Dai et al., 2021).
> > To verify that this choice does not produce unique conclusions regarding efficacy of online distillation, we do experiments on CIFAR-100 with various convex combinations of standard and knowledge distillation losses. The results presented in Table 6 of the revision confirm our main conclusions on online distillation. Furthermore, we observe that ignoring dataset labels does not result in significant degradation of student performance.
> >
> > > (7) How do you explain the gains in binary classification CIFAR100 experiments, if not for the so-called "dark knowledge"?
> >
> > In the paper we argue that performance of knowledge distillation cannot be attributed to “dark knowledge” solely, precisely because in the case of binary classification on CIFAR-100 where there is not much “dark knowledge”, one can still observe significant improvement with KD. We attribute this to the lower supervision complexity of teacher predictions compared to dataset labels (Proposition 5 and Figure 2a). Note that there are other theoretical works that suggest that the success of knowledge distillation can be partly attributed to some optimization biases, e.g., (Mohabi et al., 2020) and (Phuong et al., 2019).
> >
> > > (8) Prop.5 explains the effect of temperature on the KD generalization. How does this explain the effect of early stopping? Similarly, how does this explain the effect of teaching a weak student?
> >
> > Overall, Proposition 5 highlights that there is a trade-off between teacher accuracy, supervision complexity, and student margin with respect to teacher predictions. Early stopped teachers have worse accuracy and perhaps results in a smaller student margin, but they also have a significantly smaller supervision complexity. This provides a possible explanation of why early stopped networks are sometimes better teachers compared to fully trained networks.
> >
> > As we show, a fully trained teacher can have predictions as complex as random labels for a weak student at initialization. This low alignment of student NTK and teacher predictions can result in memorization if the student NTK stays the same during the training. To understand why offline distillation works, one needs to understand how the NTK changes after a few iterations. This is beyond Proposition 5. Nevertheless, Proposition 5 gives some clues of why online distillation is a better choice for weak students. In online distillation, the early teacher checkpoints have less complex predictions, as neural networks learn simple patterns first. These early teacher predictions align with the weak student initial NTK significantly better, allowing the student to learn these patterns in a generalizable fashion. This feature learning improves the student NTK and allows learning more complex patterns in future iterations.
> >
> > We have added some discussion on these topics in Section D.
> >
> > > (9) Why doesn't the NTK result by Arora et al (2019) directly give you the supervision complexity term? It has the form $\\boldsymbol{Y}^\\top (K^\\text{inf})^{-1} \\boldsymbol{Y}$.
> >
> > Indeed terms of the form $\\boldsymbol{Y}^T K^{-1} \\boldsymbol{Y}$ appear in the literature of kernel methods (see the extended related work of the Appendix E). Our novel contribution is in *viewing this term in the context of knowledge distillation* in order to explain some aspects of knowledge distillation and give valuable insights for improving KD methods.
> > While Theorem 4 of our work is similar to the generalization bound of Arora et al. (2019), there are several key differences.
> > * The result of Arora et al. (2019) is only for neural networks with one hidden layer, with specific parameterization and initialization, and training with full-batch gradient descent, where only the first layer parameters are trained. In contrast, our result applies to all kernel methods of form Equation (3) of the paper.
> > * We consider the empirical NTK, rather than the expected NTK.
> > * Our proof is much shorter and doesn’t rely on the specifics of the training process.

---

### Official Review · Reviewer_jR2a · 2022-10-25

**Confidence:** 3
**Correctness:** 4
**Technical Novelty And Significance:** 3
**Empirical Novelty And Significance:** 2
**Recommendation:** 6

**Clarity, Quality, Novelty And Reproducibility:**

The paper's theoretical quality is high, but can be improved in terms of clarity particularly in explaining the paragraph "Implications for neural classifiers". The theoretical understanding behind effectiveness of distillation seems to be novel.

**Strength And Weaknesses:**

Strengths -

1. The paper presents a strong theoretical understanding behind the role of knowledge distillation particularly the effect of using smoother labels and temperature smoothing for the teacher.

2. The proposed Online distillation achieves strong empirical improvements over the baseline Offline distillation.

Weakness -

1. One thing which is unclear is for the experiments on distillation do the authors also train with the dataset labels or only train with the teacher targets? If it is the latter, can authors also show results comparing Online/Offline distillation when the dataset labels are also used for training. I can see that the dataset labels was excluded for the theoretical analysis, but for a fair comparison with the way distillation is used generally the dataset labels should be included in the training of the student model.

2. While in Figure 2c) the paper compares the Supervision Complexity with  an average teacher, showing that the Supervision Complexity of  Average teacher is higher than that of Online teacher, I could not see any comparison between Online distillation and distillation done with an Average teacher. If the Supervision complexity is higher, does it mean that the Average teacher will perform better than the Online distillation?

3. For the results showing high correlation between NTK similarity and the test accuracy, can the authors do further experiments to validate this. For instance, does a weaker student model (LeNet) have better similarity than a stronger student (ResNet-20)? Or does the student model trained with augmentations have better NTK similarity with the teacher than the student model trained without augmentations.

4. In table 1, Online distilation from ResNet-56 to ResNet-20 performs worse than Offline distillation. Can the authors also show the student and teacher trajectories for this case (Figure 1) to better understand what is happening for this case?

**Summary Of The Paper:**

The paper proposes to view knowledge distillation from the Supervision Complexity angle, or how easy/hard is it for the student model to learn the targets. One reason why distillation has been thought to be helpful is that the soft labels additional provide information about class similarities which the one-hot labels do not provide. However, distillation has been also shown to be useful for binary classification problems where there is a limited class similarity information. The paper thus looks at other alternatives to explaining distillation, especially the reasoning behind  using soft-labels from teacher and that of using temperature smoothing. They come up with a notion of Supervision Complexity which intuitively determines how easy it is for the teacher to learn from the targets.  They theoretically show that for kernel based binary classifiers, the generalization is bounded by Supervision Complexity and then later extend it to multi-class kernel based classifiers. The Supervision complexity can be reduced by reducing the scale of the targets or by aligning the targets with the eigenvectors of the kernel. Then they finally extend their analysis for neural networks by treating them as corresponding linearized neural networks.

For distillation, the theoretical analysis reveals that for a binary classification setting, the student risk is bounded by the sum of Supervision Complexity and the Student's margin loss with teacher targets. Thus, making the targets softer in distillation reduces Supervision Complexity and correspondingly helps in better generalization. However, increasing the temperature a lot (and thus making targets softer) would also affect the margin loss and thus the temperature smoothing for distillation needs to happen carefully.

The authors also show that in the initial stages of training, the Supervision Complexity from soft-targets can be same as that of one-hot labels particularly if the gap between student and teacher is high. Thus to remedy this the authors propose (Online distillation)gradually increasing the complexity of the teacher for distillation, whereby the teacher checkpoints used for distillation are gradually increased. Empirically it is shown that Online distillation works better than offline distillation for several datasets.

Finally the paper mentions that the test accuracy of the student is highly correlated with the similarity between the student and the teacher NTK matrices (computed over a batch of data), and show that the similarity is indeed higher for Online distillation as compared to Offline distillation.

**Summary Of The Review:**

While I like the general direction pursued in the paper around connecting knowledge distillation with Supervision complexity, I have some concerns with the experimental settings which I have mentioned in the weakness section. I am leaning towards weak accept for now, but will update my ratings if my concerns are addressed.

---

> ### Author Response · Authors · 2022-11-11
> **Response to Reviewer jR2a (Part I)**
>
> We would like to thank the reviewer for the precise reading of our work.
>
> > 1. One thing which is unclear is for the experiments on distillation do the authors also train with the dataset labels or only train with the teacher targets? If it is the latter, can authors also show results comparing Online/Offline distillation when the dataset labels are also used for training. I can see that the dataset labels was excluded for the theoretical analysis, but for a fair comparison with the way distillation is used generally the dataset labels should be included in the training of the student model.
>
> In all knowledge distillation experiments reported in this work, we don't use dataset labels as an additional source of supervision. We noted this following Equation 2 of the paper, but have now made this more explicit in the experiments section.
>
> We did not use dataset labels to minimize differences between the theory and experiments. Such a choice has also been made in prior theoretical analyses, e.g., (Menon et al., 2021) and (Dai et al., 2021).
> To verify that this choice does not produce unique conclusions regarding efficacy of online distillation, we do experiments on CIFAR-100 with various convex combinations of standard and knowledge distillation losses. The results presented in Table 6 of the revision confirm our main conclusions on online distillation. Furthermore, we observe that ignoring dataset labels does not result in significant degradation of student performance.
>
>
> > 2. While in Figure 2c) the paper compares the Supervision Complexity with an average teacher, showing that the Supervision Complexity of Average teacher is higher than that of Online teacher, I could not see any comparison between Online distillation and distillation done with an Average teacher. If the Supervision complexity is higher, does it mean that the Average teacher will perform better than the Online distillation?
>
> We were mainly referring to the work of Ren et al. (2022). A similar effect has been observed in semi-supervised learning (see for example Tarvainen and Valpola (2017)). Implementing knowledge distillation with an average teacher is computationally expensive as one needs to run inference for 10 teachers at each iteration. Unfortunately, the caching solution does not apply in our setting, as we use data augmentations.
>
> > 3. For the results showing high correlation between NTK similarity and the test accuracy, can the authors do further experiments to validate this. For instance, does a weaker student model (LeNet) have better similarity than a stronger student (ResNet-20)? Or does the student model trained with augmentations have better NTK similarity with the teacher than the student model trained without augmentations?
>
> We would like to draw the reviewer’s attention to Figure 9 (Appendix), where we present additional evidence of the correlation between NTK similarity and test accuracy. We have added results for two more datasets, teacher, and student combinations.
>
> In general, we observe that stronger the student the better it matches the NTK of its teacher. For example, ResNet-20 has a higher NTK similarity with the ResNet-56 teacher compared to LeNet-5x8 (see Fig. 3(a) and Fig. 3(b)), and VGG-16 has a higher NTK similarity with the MobileNet-V3-125 teacher compared to MobileNet-V3-35 (see Fig. 9(g) and Fig. 9(k)).
>
> _References_
>
> [1] Tarvainen, A. and Valpola, H., 2017. Mean teachers are better role models: Weight-averaged consistency targets improve semi-supervised deep learning results. Advances in neural information processing systems, 30.

---

> > ### Author Response · Authors · 2022-11-11
> > **Response to Reviewer jR2a (Part II)**
> >
> > > 4. In table 1, Online distillation from ResNet-56 to ResNet-20 performs worse than Offline distillation. Can the authors also show the student and teacher trajectories for this case (Figure 1) to better understand what is happening for this case?
> >
> > In general, we observed that online KD increases student’s test accuracy slower, especially when the temperature is high. In early epochs of online KD, teacher predictions have high uncertainty, which is further increased by temperature scaling. This results in small gradient updates and slow learning. We combat the effect of temperature by scaling the knowledge distillation loss by $\\tau^2$ in the case of cross-entropy loss (as in Hinton et al. (2015)) and by $\\tau$ in the case of mean squared error loss. Given that we train students for the same total number of epochs in both offline and online distillation, it is possible to observe online distillation performing worse than online distillation. We see a few ways to avoid this:
> > * increasing the training time,
> > * using smaller temperature in early epochs of online distillation,
> > * increasing the learning rate in early epochs of online distillation.
> >
> > We did not pursue this direction to not deviate from our main goal of highlighting the role of supervision complexity in knowledge distillation.
> >
> > We are working to add a few training curve plots to demonstrate that online KD increases student accuracy slower.
> >
> > > 5. The paper's theoretical quality is high, but can be improved in terms of clarity particularly in explaining the paragraph "Implications for neural classifiers".
> >
> > We have made some updates to the paragraph “Implications for neural networks” to make its connection with generalization bounds of Theorems 3 and 4 more clear.

---

### Author Response · Authors · 2022-11-11
**To All Reviewers**

We would like to thank all the reviewers for useful comments and feedback. We appreciate the consensus among the reviewers on the significance of the theoretical contributions of our work.

A common point raised by all reviewers is that in some cases online knowledge distillation does not outperform offline knowledge distillation. We agree with this. In the case of ResNet-56 teacher and ResNet-20 student, there is not much room to improve as the student’s performance approaches the teacher's performance in offline distillation. In the case of ResNet-110 teacher and ResNet-20 student we are not sure what the reason is. In the remaining cases there are large architectural differences between the student and teacher, and we observe significant performance gains with online distillation (e.g., CIFAR-100 with LeNet-5x8 student or Tiny-ImageNet with VGG-16 student).

Overall, we wish to reiterate that the main goal of our work is to explain some aspects of knowledge distillation from the perspective of supervision complexity. We have shown that when the student network has weak inductive biases, online distillation can lead to significant performance gains, supporting our theoretical observations on the role of supervision complexity. Making online distillation memory efficient or improving its performance further was not our goal. This is an interesting open problem that can be addressed by for example picking teacher checkpoints more strategically, having a temperature schedule that adapts to supervision complexity changes, and training the student longer.

We have uploaded an updated manuscript with the following changes.
* Additional experiments on CIFAR-100 to demonstrate that mixing knowledge distillation and standard supervised classification losses (Table 6), or picking a different temperature parameter (Table 7) do not affect our main conclusions regarding the efficacy of online distillation.
* A new experiment where the frequency of teacher checkpointing is varied (Figure 8).
* Two more plots showing high correlation between teacher-student NTK similarity and student test accuracy (Figure 9).
* A few notes on early stopped teachers and teaching students with weak inductive biases (Section D).
* More details in the “Implications for neural classifiers” part of Section 2.2.
* Due to the space constraints we have moved the future work section to the “Expanded Discussion on Related Work” section (Appendix E).

---

### Author Response · Authors · 2022-12-07
**To All Reviewers**

We again thank the reviewers for providing valuable feedback on our submission. We have posted detailed responses to the comments and questions. Given that the discussion period will end soon, we would like to check if the reviewers found our responses satisfactory. We would be happy to address any remaining comments and questions.

---

### Decision · Program_Chairs · 2023-01-20

**Decision:**

Accept: poster

**Justification For Why Not Higher Score:**

The experimental results are limited.

**Justification For Why Not Lower Score:**

The theoretical insight is very interesting and should be interesting to the majority of ICLR audience.

**Metareview: Summary, Strengths And Weaknesses:**

It is a novel paper that explains the generalization of KD using a new theory: supervision complexity. This theory not only explains some popular tricks but also suggests that online distillation would be a better direction.

Strength:
1. New theoretical insight about KD.
2. Online KD is a promising application of the theory.

Weakness:
1. Presentation is a little bit dry. Authors may want to provide some informal analysis in the main paper and move the technical proof in Appendix.
2. Experimental improvement over baselines is limited.

**Note From Pc:**

if the above contains the word "oral" or "spotlight" please see: "oral" presentation means -> notable-top-5% and "spotlight" means -> notable-top-25%. As stated in our emails, we are disassociating presentation type from AC recommendations